



# Hydrography and circulation in the vicinity of the central Getz Ice Shelf: two years of mooring observations

Vår Dundas[1], Elin Darelius[1], Kjersti Daae[1], Nadine Steiger[1], Yoshihiro Nakayama[2], and Tae-Wan Kim[3]

[1]Geophysical Institute, University of Bergen and the Bjerknes Centre for Climate Research, Norway
[2]Institute of Low Temperature Science, Hokkaido University, Sapporo, Japan
[3]Korea Polar Research Institute, Incheon, South Korea

**Correspondence:** Vår Dundas (var.dundas@uib.no)

**Abstract.**

Ice shelves in the Amundsen Sea are thinning rapidly as ocean currents bring warm water into the cavities beneath the floating ice. While the reported melt rates for the Getz ice shelf are comparatively low for the region, its size makes it one of the largest freshwater sources around Antarctica, with potential consequences for e.g. bottom water formation downstream.

Here, we use two year long mooring records (2016 – 2018) and 16 year long regional model simulations to describe, for the first time, the hydrography and circulation in the vicinity of the ice front between Siple and Carney Island. We find that, throughout the mooring record, temperatures in the trough remain below $0.15°C$, more than $1°C$ lower than in the neighboring Siple and Dotson Trough, and we observe a mean current ($0.03$ m s$^{-1}$) directed towards the ice front. The variability in the heat transport towards the ice front appears to be governed by wind stress over the Amundsen Sea Polynya region, potentially through

interactions with the coastal current, although this hypothesis could not be confirmed by the numerical model. The model simulations suggest that the heat content in the trough during the observed period was lower than normal, due to anomalously low summertime sea ice concentration and weak winds.

## 1   Introduction

Getz Ice Shelf (GIS) in the western Amundsen Sea is among Antarctica's primary sources of meltwater (Rignot et al., 2013) and main contributors of ice shelf volume loss (Paolo et al., 2015), with basal melt rates approaching 5 m yr$^{-1}$ (Rignot et al., 2013). Despite the stabilizing buttressing effect of islands that separate its ice fronts (Fig. 1a, e.g. Heywood et al., 2016; Shepherd et al., 2018; Jacobs et al., 2013; Dupont and Alley, 2005), the GIS grounding line is retreating (Shepherd et al., 2018), potentially influencing the stability of GIS. The high melt rates enhance the meltwater fraction transported from GIS

and westward to the Ross Sea (Nakayama et al., 2014a, 2020), connecting changes in the GIS region to the global climate: More meltwater in the Ross Sea is suggested to affect the Antarctic Bottom Water production and the global thermohaline





circulation, and consequently the global ocean overturning (Nakayama et al., 2014a, b). Despite this connection between the ocean-driven melt of GIS and the global climate, the area is severely undersampled.

**Figure 1.** a) Map of the Amundsen Sea with bathymetry (color scale) (IBCSO, Arndt et al. 2013) and ice shelf (grey) (Bedmap2, Fretwell et al. 2013). The location of the study region within Antarctica is indicated by a red box in the inset. The moorings $GC_6$ (red star), $GW_6$, $GW_{6F}$, and S1 (red dots) are marked, with arrows denoting mean current averaged over depth and time. The velocity scale is given in the lower left corner. Abbreviations are SI: Siple Island, CI: Carney Island, ASP: Amundsen Sea Polynya, and DG-Trough: Dotson-Getz Trough. The outline of the ASP is based on data from January 2011 (Yager et al., 2012). b) Detailed map of the $GC_6$-trough. CTD-stations from ship-borne surveys (yellow stars) are shown in a) and b). CTD stations from an instrumented seal are marked with orange dots. c) The mean wind field from 2001 to 2018 (ERA 5, color scale and purple arrows, see black arrow for scale) with mean zero-contours during the 17 year period (cyan), summer (red), and winter (blue). The thin meridional red lines indicate the meridional band used for estimating the latitude of the zero-contour north of $GC_6$. The black contour is the 900 m isobaths. The red SB- (a), ASP, and East-boxes (c) indicate regions used for averaging ocean surface stress. The SB-box is also used for estimating cumulative Ekman pumping anomalies.

The presence of warm and dense Circumpolar Deep Water (CDW, core temperature of $2°C$, Heywood et al. 2016) and

its slightly colder modified version (mCDW) on the continental shelf is the main cause of the high basal melt rates in the



Amundsen Sea (Rignot et al., 2019). CDW is found just off-shelf of the continental shelf break, a characteristic specific to West Antarctica (e.g. Holland et al., 2020). However, the Getz region's regional differences are large (Jacobs et al., 2013). GIS spans roughly 650 km along the coast (Jacobs et al., 2013) and is sectioned into several ice shelf fronts by six islands (Assmann et al., 2019). The differences in local bathymetry, together with variations in the regional wind field, the Antarctic
Slope Front (ASF), an along-slope undercurrent, and the depth of the thermocline relative to the ice shelf grounding lines cause large differences in ice thickness change within the Amundsen Sea (Paolo et al., 2015; Shepherd et al., 2018). Together, these aspects influence whether the CDW from the deep ocean is allowed onto the continental shelf and whether it reaches the ice shelf bases to the south. Different combinations of the mechanisms that admit on-shelf heat transport dominate at different locations, and consequently, each GIS frontal region needs to be studied separately.

In this paper, we present the first mooring record, $GC_6$ (Getz Central, 650 m, 2016-2018, Fig. 1a), near the GIS front between Siple and Carney Islands. This trough (referred to as the $GC_6$-Trough hereafter) has until now been overlooked, compared to the neighboring Siple Trough in the west, and the Dotson-Getz Trough in the east, which have recently received attention (Assmann et al., 2019; Wåhlin et al., 2020; Steiger et al., 2021; Wåhlin et al., 2010, 2013; Kalén et al., 2016). Historically, few observations exist from the $GC_6$-Trough, and consequently, our new mooring observations enable a first description of the
oceanography in the trough, beyond previous descriptions based on snapshot CTDs (Jacobs et al., 2013).

    The surface winds drive a wide range of processes that affect on-shelf heat transport, such as Ekman pumping at the shelf break (Assmann et al., 2019), on-shelf current variability (Wåhlin et al., 2013), and the on-shelf flow of CDW (Thoma et al., 2008). The winds are predominantly westward along the coast and eastward north of the shelf break. The latitude where these zonal winds shift direction, the "zero-contour", generally migrates northward in summer and southward in winter (Assmann
et al., 2013). While the eastern part of the shelf break experiences a seasonal shift in zonal winds, the western part is usually affected by easterlies year-round due to its higher latitude.

    Variations in the ASF, the associated Antarctic Slope Current (ASC), and its undercurrent are related to these wind patterns (Dotto et al., 2019). The ASF is a wind-driven frontal system at the continental shelf break (Jacobs, 1991). Its associated sharp thermocline and downward-sloping isotherms from north to south impede the flow of CDW across the shelf break (e.g.,
Thompson et al., 2018), and consequently regulate the amount of heat on the continental shelf. Easterlies sharpen the ASF through downward Ekman pumping (e.g., Thompson et al., 2018), while surface stratification dampens this effect and relaxes the ASF (Daae et al., 2017). The eastward undercurrent is maintained by the horizontal density gradients across the ASF (e.g. Smedsrud et al., 2006; Chavanne et al., 2010). When the isopycnals of the ASF are steep enough to sustain the undercurrent and relaxed enough to admit water below the thermocline over the trough sills, this undercurrent current brings warm water
directly into troughs (Walker et al., 2013; Assmann et al., 2013).

    While the adjacent Siple and Dotson-Getz Troughs cross-cut the continental shelf (Fig. 1a), the $GC_6$-Trough does not reach the shelf break, although it is $\sim$ 1000 m deep at the ice front (Fig. 2c). The warm along-slope undercurrent therefore often crosses the deep Siple and Dotson-Getz Trough sills ($\sim$ 570 and $\sim$ 500 m deep), but not the shallower $GC_6$-Trough's sill ($\sim$ 460 m deep). However, although the shallow sill-region north of $GC_6$ is also likely accompanied by a relatively deep





thermocline (Jacobs et al., 2012), water roughly $2°$C above freezing was observed in the $GC_6$-Trough by a snapshot CTD in
2007 (Jacobs et al., 2013). Unmodified CDW is also present directly north of the shelf break (Fig. 3b).

We describe the observed hydrography and currents at $GC_6$ based on the mooring records and compare the observations
to those from neighboring troughs (Siple and Dotson-Getz). We discuss their variability and possible drivers, with a specific
focus on forcing by ocean surface stress ($\tau$). To set the two years of mooring observations in a broader temporal and spatial

perspective and compensate for the sparse observational data coverage, we investigate output from a high-resolution regional
model run (Nakayama et al., 2018) and include historical CTD data from the trough.

## 2   Data and methods

### 2.1   Observational data

The mooring $GC_6$ was deployed during the Amundsen Sea Expedition 2015 – 2016 (ANA06B) and collected data from 30

January 2016 to 31 January 2018. It was located at $648$ m depth on the eastern slope of the $GC_6$-Trough (123.6W, 73.7S),
about 30 km north of one of GIS's ice fronts (Fig. 1a,b). $GC_6$ recorded temperature, salinity, pressure (SBE56 and SBE37
from Seabird Electronics), and current velocity (RDI ADCP, 150kHz, for instrument levels see Fig. 2b). The data are corrected
for magnetic declination, outliers are removed, and the ADCP data are processed with the RDI software following standard
procedures. We use hourly and daily averaged data of all variables. We follow TEOS-10 (IOC et al., 2010) and present the

hydrographic data as absolute salinity ($S_A$) and conservative temperature ($\Theta$) with $\delta S_A$ taken from version 3.6 of McDougall
et al. (2012) database.

We rotate the coordinate system to follow the mean flow direction ($\sim 174°$) at $GC_6$ (Fig. 2c,d), which roughly corresponds
to the along-trough direction. A positive along-trough current (AT in Fig. 2c) is directed toward the ice shelf (south-southeast).
Correspondingly, we define a cross-trough direction toward east-northeast (CT in Fig. 2c). We approximate heat content at $GC_6$

as a weighted sum of each level of temperature measurements, using density $\rho = 1028$ kg m$^{-3}$ (Dotto et al., 2019) and specific
heat $cp = 3985$ J Kg$^{-1}$ K$^{-1}$. We approximate heat transport as the heat flowing past the mooring. For both approximations,
we use the temperature relative to in situ freezing temperature. These estimations give an upper limit of the heat available to
potentially melt ice, assuming that all the water containing this heat reaches the ice shelf base unaltered. Since measurements
are only available along one axis, the heat content and the heat transport we present have units J m$^{-2}$ and W m$^{-1}$.

In addition to the new observations from $GC_6$, we use mooring records from the neighboring Siple Trough ($GW_6$ and $GW_{6F}$
from 2016-2018, Assmann et al., 2019) and the Dotson-Getz Trough (S1 from 2010-2014, Wåhlin et al., 2013; Kalén et al.,
2016; Arneborg et al., 2012). We include all existing CTD-profiles (12 in total) between the $GC_6$-region and the shelf break,
which were obtained during cruises with *N.B. Palmer* (1994, 2000, 2007) and *Araon* (2016, 2018) (Fig. 1a.). One instrumented
seal (the MEOP project,  Mcintyre et al., 2017) visited the mooring site and provided 23 CTD profiles between 12 and 16

March 2014. For bathymetry, we use the International Bathymetric Chart of the Southern Ocean Version 1.0 (IBCSO, Arndt
et al. 2013). Multibeam recordings taken under deployment of $GC_6$ (Lee, 2016) reveal inaccuracies in the bathymetry presented





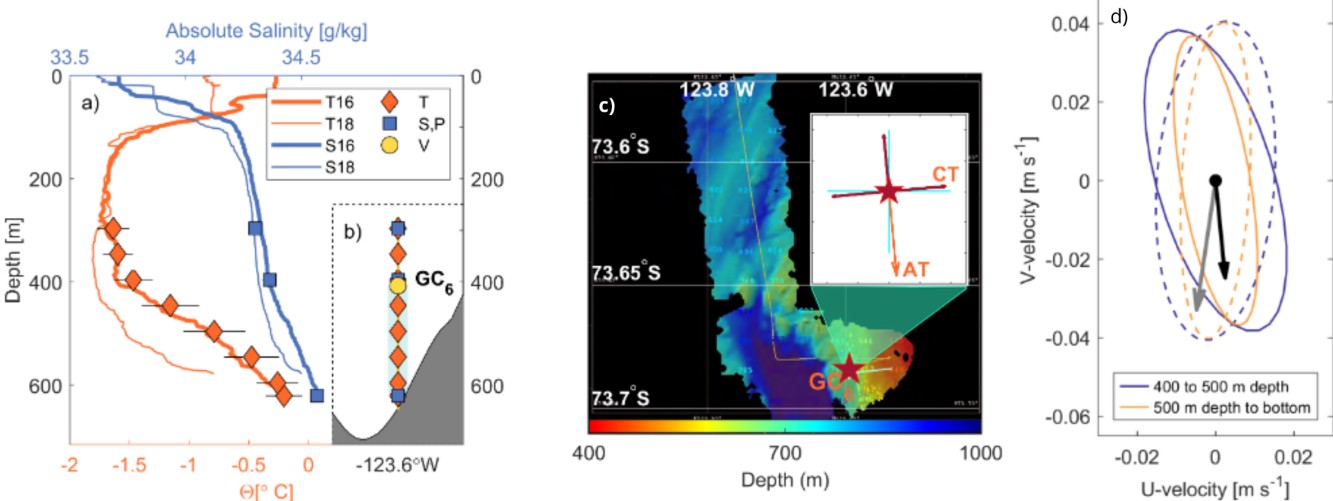

**Figure 2.** a) Profiles of conservative temperature (orange) and absolute salinity (blue) obtained from CTD casts at deployment (thick lines) and recovery (thin lines) of mooring $GC_6$. Mean temperature (orange diamonds) and salinity (blue squares) recorded by each instrument on $GC_6$ with standard deviation (black lines) is marked. The standard deviation for salinity is too small to be seen outside the blue boxes. b) Instrument levels for temperature, salinity, pressure, and velocity (with the range in light blue) on $GC_6$. The grey patch shows approximate bathymetry (based on IBCSO). c) Bathymetry in the trough from multibeam recordings (adapted from Lee 2016) with the location of $GC_6$ marked by the red star. The rotation of the coordinate system at $GC_6$ from cartesian coordinates (cyan lines) to coordinates based on the mean current direction (red lines) is indicated in the inset. The orange arrow shows the mean current direction (AT: along-trough), while CT denotes the cross-trough direction. d) Velocity variance ellipses from 407 m (ADCP instrument depth) to 500 m depth (blue) and below 500 m depth (orange). Solid lines for the mooring and dashed lines for the regional model's daily output. The total mean velocities for observations and model are indicated by black and grey dots with arrows, respectively.

in the IBCSO dataset in the $GC_6$ region – the bathymetry is rougher and steeper than the IBCSO bathymetry indicates (Fig. 1b and 2c).

## 2.2 Regional model data

To complement the observational data, we use results from a regional model run (Nakayama et al., 2018, 2019) using the Amundsen and Bellingshausen Sea configuration of MITgcm. The model has a nominal horizontal grid spacing of about $1/12°$, and the lateral boundary conditions are the ECCO LLC270 optimization. Its atmospheric forcing is from the ERA-Interim reanalysis (Dee et al., 2011). The model bathymetry is based on IBCSO.

Comparing the observed and modeled daily temperature and current at the $GC_6$ mooring site (Appendix A) we draw two
main conclusions: i) The variability in the depth of the $-1°C$ isotherm compares relatively well (r= 0.31, bandpass filter from 8 days to 10 months, Fig. A1b and A1c). However, the average depth of isotherms is shallower in the model ($368 \pm 27$ m depth, during the period that overlaps with the mooring period) than in the observations ($461 \pm 28$ m). ii) The modeled average velocities agree well with observations (Fig. 2d), but the current variability at $GC_6$ is not captured by the model (Fig. A1e). The model has proved reliable in simulating the undercurrent, the flow of warm water across the shelf break into the cross-



cutting troughs, and general conditions in the Eastern Amundsen Sea (Nakayama et al., 2018, 2019). Therefore, we rely on its large-scale currents and temperature variability. We note that trough openings are generally deeper in the regional model than in the IBCSO bathymetry. This might explain differences between model results and observations, such as the overestimated thickness of the warm deep layer at $GC_6$.

We use daily (available Jan. 2016 to Sept. 2017) and monthly (available Jan. 2001 to Sept. 2017) means of temperature and

current velocity of the model output. The model was initially run for another project, and consequently, the run ends five months earlier than the $GC_6$ record. We use the monthly model output to look into low-frequency processes, and the daily output to look further into results based on the $GC_6$ mooring observations. We select a location $GC_{6\_mod}$ (Fig. 8a) representative of $GC_6$'s depth and location relative to the trough's bathymetry.

### 2.3  Atmospheric reanalysis data

We use reanalysis output of 10 m wind and sea ice concentration (SIC) from ERA 5 (Hersbach et al., 2020), and the Polar Pathfinder Daily 25 km EASE-Grid Sea Ice Motion Vectors, Version 4 (Tschudi et al., 2019) to estimate $\tau$ following Dotto et al. (2018) (referred to as $\tau_{D18}$ hereafter, Eq. B2a). This estimation assumes a motionless ocean and constant drag coefficient. Values are missing along the coast in the NSIDC data set, but since the sea-ice movement is expected to be small along the coast in winter, we assume little loss of information. We use SIC from ERA 5 for consistency with the wind velocities. The

cumulative Ekman pumping anomaly (wEK) is calculated as described in Appendix B. The monthly mean meridional location of the zero contour is estimated over a meridional band over the $GC_6$ mooring location (meridional red lines in Fig. 1c). For analysis involving model output, we use daily instantaneous surface stress reanalysis output from ERA-Interim (referred to as $\tau_{ERA-I}$ hereafter) since this is used to force the regional model. Based on SIC we define summer (Dec-Apr) and winter (May-Nov).

### 2.4  Statistical methods

To estimate the temporal evolution of correlation we use a 100-day moving window with 10 days overlap. All correlation values are significant on the 95% level, with significance calculated following Sciremammano (1979). We allow a maximum lag of 7 days for correlation analyses involving currents, and 30 days for correlation involving temperature and isotherm variability, which encompasses rapid barotropic responses but leaves out slow advective signals. The mooring record length and low

velocities in the $GC_6$-Trough yield too few degrees of freedom to allow for a lag on the order of the advection timescale of roughly 3 months from the shelf break to $GC_6$ Sciremammano (1979).

To remove diurnal and seasonal signals from the mooring observations and model output we use two Butterworth filters. For the observations, we apply a bandpass filter from 8 days to 10 months ($BP_{8D-10M}$). For the model output, we remove estimated seasonal cycles based on the 16-year long monthly time-series and lowpass filter at 8 days ($LP_{8D}$).

In sect. 3.3.4 we use the following procedure to produce correlation maps between the zonal ocean surface stress, $\tau_{ERA-I}$, and the modeled currents on the continental shelf: The currents are separated into deep currents (depth-averaged below the $0°C$-isotherm) and surface currents (depth-averaged above 100 m). In each grid cell, the coordinate system is aligned with the





vector-averaged current direction. The component of the current aligned with the (spatially varying) mean current direction is then correlated with zonal $\tau_{\text{ERA-I}}$ averaged over a fixed region.

## 140 3 Results

We present hydrography, currents, heat content, and heat transport based on the mooring observations from the $GC_6$-Trough and investigate how these variables are influenced by regional atmospheric forcing primarily through correlation analysis. To set the mooring period in a larger temporal and spatial perspective, we assess the variability in surface forcing from 2001 to 2017 and use the regional model output to further investigate the connection between atmospheric forcing and isotherm depth
and the currents at $GC_6$.

### 3.1 Observations from mooring $GC_6$: 2016-2018

#### 3.1.1 Hydrography

A bottom layer of relatively warm modified Circumpolar Deep Water (mCDW), a mixture between Winter Water (WW, $-1.8°C$) and CDW is present at $GC_6$ throughout the mooring period (Fig. 3c and 4a). This layer is overlain by WW which
ventilates down to $\sim 450$ m depth in October 2016 and October/November 2017 (Fig. 4a). The depth-averaged temperature and salinity at $GC_6$ are $-0.95°C$ and $34.41$ g/kg, respectively. The maximum recorded temperature at $GC_6$ is $0.13°C$ - more than two degrees above freezing, but almost $1.5°C$ lower than the maximum temperatures at $GW_6$ and S1 (Fig. 3). The temperature records show no apparent seasonal signal in the thickness or properties of the mCDW layer. The highest temperatures ($\Theta > 0°C$) are observed at the end of 2016 when the thickness of the mCDW layer is at its maximum, followed by a gradual
thinning of the mCDW layer (Fig. 4a).

Several sudden cooling events in 2016 at $GC_6$ correspond to cold events at $GW_{6F}$ (Fig. 4a,b,d, green diamonds) which were triggered remotely by strong easterly winds that induced coastal Ekman downwelling, pushing the WW to the bottom (Steiger et al., 2021). Although the signal fades with distance from the ice shelf front (Steiger et al., 2021), the events are registered at $GC_6$, which was 30 km from the ice front. In 2017 the events are weaker and the signal less prominent in the colder bottom
temperatures at $GC_6$, similarly to at $GW_{6F}$.

When a watermass is in contact with and melts glacial ice, its $\Theta S_A$-properties will evolve along the "Gade-line" in $\Theta S_A$-space (Gade, 1979). Such alignment is observed in the Dotson-Getz Trough at mooring S1 (Fig. 3d, at $\Theta = 0.1$, $S_A = 34.6$) but is not observed in the Siple- nor the $GW_6$-Troughs, including the $GC_6$ mooring, the CTD casts, and seal dives (Fig. 3b,c, Fig. 1a for CTD-locations).






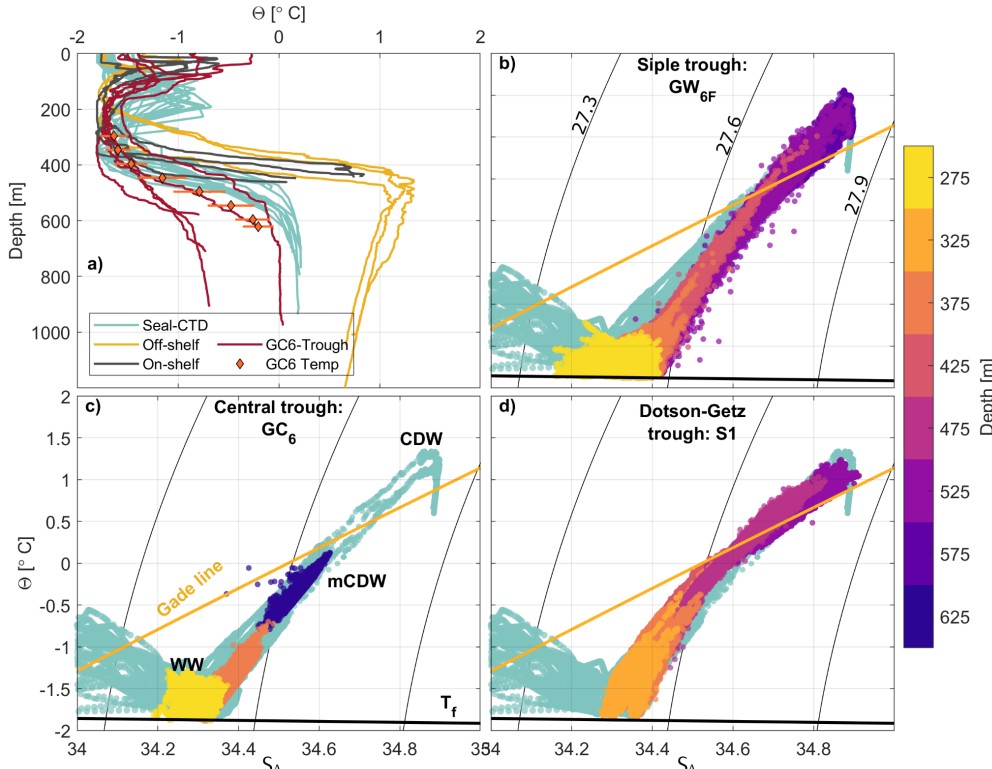

**Figure 3.** a) Temperature profiles taken by ship (red, black, yellow) and seal (turquoise, see Fig. 1a,b for locations). Orange diamonds and lines show the mean temperature and standard deviation recorded by each instrument on GC6. $\Theta S_A$-diagrams for b) $GW_{6F}$ in the Siple trough, c) $GC_6$ in the $GC_6$-Trough between Siple and Carney Islands, and d) S1 in the Dotson-Getz Trough (see Fig. 1a for locations), color-coded by the depth of moored instruments, and with the CTD-stations and seal dives from Fig. 1a marked in turquoise in the background. Four outliers are removed from S1. The $\sigma$ density contours (thin, black lines), the Gade-line (yellow), surface freezing temperature $T_f$ (thick, black line), and characteristic water masses are labeled (WW: Winter Water, mCDW: modified Circumpolar Deep Water, and CDW: circumpolar deep water).

### 3.1.2 Currents

The current's magnitude and variability are highest in the along-trough direction (Fig. 2d, solid lines) with an average velocity and standard deviation of $3 \pm 5$ cm s$^{-1}$. It is directed toward the ice shelf (Fig. 1a and 4d) and is nearly depth-independent (barotropic) over the observed depth ($407 - 615$ m). There is a tendency for a stronger current away from the ice front (negative values in Fig. 4b) during summer 2017. The variability in the depth-averaged along-trough velocity and the depth-averaged temperature from the four bottom sensors (referred to as the mCDW-temperature hereafter) are negatively correlated for most of 2016 (r$= -0.50$, lag$= -2.5$ days, temperature leading), and positively correlated for most of 2017 (r$= 0.52$, lag$= 2$ days) when applying BP$_{8D-10M}$ (not shown). Consequently, in 2016, periods of high temperatures coincide with strong currents away from the ice shelf, while in 2017 periods of high temperatures coincide with strong currents toward the ice shelf.

### 3.1.3 Heat content and heat transport

The average heat content relative to the in situ freezing point at $GC_6$ is $1.7 \pm 0.2$ GJ m$^{-2}$. This value is lower than the $GW_6$-mooring in the Siple-Trough even though $GC_6$ was moored at a greater depth (650 m vs. 600 m), where the water is typically



**Figure 4.** Daily mean records from $GC_6$ showing a) conservative temperature, and b) depth-averaged along-trough (AT) velocity at $GC_6$ (red) and zonal $\tau_{D18}$ averaged over the SB-box (blue). Positive values denote flow toward the ice shelf and eastward stress, respectively. The $-1.8°C$ (cyan), $-0.5°C$ (white), and $0°C$ (magenta) contours are highlighted in a), and the measurement depths are shown (black diamonds) in a). c) Heat content (HC) at $GC_6$ (purple) and $GW_6$ (dashed purple), and heat transport (HT) at $GC_6$ (green) and $GW_6$ (dashed green), all $LP_{8D}$. d) Cumulative Ekman pumping anomaly in the SB-box (filled blue) and SIC over $GC_6$ (black), with their mean seasonal cycles (2001-2018) in dark blue and grey, respectively. e) Estimated monthly mean location of the zero-contour in a meridional band spanning the zonal extent of the SB-box. The filled orange area indicates westward winds and white indicates eastward winds. The thin orange and grey lines indicate the mean monthly position of the zero-contour from 2001 to 2018, and the latitude of the shelf-break north of $GC_6$, respectively. Time is given as MM/YY. The green diamonds (a,b,d) and lines (d) mark the strong cooling events observed at $GW_{6F}$ (Steiger et al., 2021).

warmer. The heat content at $GC_6$ is generally higher in 2016 than in 2017, with a maximum in October 2016 (2.4 GJ m$^{-2}$)





and the minimum in January 2018 (1 GJ m$^{-2}$). This corresponds well with the variability of the mCWD layer which is also thickest in October 2016 (230 m, based on the $-0.5°C$ isotherm) and thinnest at the end of the mooring time-series.

     The variability in heat transport is dominated by current variability. The current past GC$_6$, and hence the heat transport, is directed toward the ice shelf 78% of the time (daily means), on average bringing $45 \pm 64$ MW m$^{-1}$ toward the shelf.

### 3.2   Atmospheric forcing

The wind field, wEK, and SIC exhibit large differences between 2016 and 2017 (Fig. 4d,e). First, the broad eastern Amundsen Sea shelf break region was dominated by westerlies during two consecutive years (2015 and 2016) since the zero-contours during summers 2015 and 2016 were shifted anomalously far south both north of the GC$_6$-Trough (Fig. 4e) and over the Amundsen Sea continental shelf as a whole (not shown). Consequently, the usual interruption of persistent summertime easterlies did not occur. Second, wEK deviates from its typical seasonal cycle (Fig. 4d), and values are positive throughout most of mid-2015

through 2016 (Fig. 7a). Lastly, summertime SIC is lower than the 2001-2018 mean over the GC$_6$-Trough in both 2016 and 2017 (Fig. 4d and 7a,b), and 2017 stands out with four ice-free months. Both the surface winds and the SIC influence $\tau$ and consequently the variability of the ocean currents and temperature.

### 3.2.1   Ocean surface stress-driven variability of the along-trough current and bottom temperature at GC$_6$

$\tau$ is an essential driver of heat content variability near several ice fronts in the Amundsen Sea. However, while $\tau$ in the continental shelf break region specifically determine heat content and heat fluxes in the vicinity of the ice front where cross-cutting troughs connect the sill to the ice front (Assmann et al., 2019; Dotto et al., 2019), the heat content and transport in the GC$_6$-trough particularly respond to $\tau$ on the continental shelf.

     The strongest correlation between the along-trough current at GC$_6$ and zonal $\tau_{D18}$ is found during winter over a region that

roughly extends from the Amundsen Sea Polynya (ASP, Fig. 1a) and northward to the shelf break (ASP-box: cyan box in Fig. 5a, r$= -0.49$, lag$= 0$ days, BP$_{8D-10M}$). The negative sign indicates that strong westward $\tau_{D18}$ enhances the along-trough current toward the ice front. During summer, the overall correlation becomes positive, and the high-correlation region shifts west of Siple Island (r$= 0.45$, lag$= 0$ days, grey dashed contour Fig. 5a).

     For mCDW-temperature, the maximum correlation with $\tau_{D18}$ is also found during winter, but in an area further east, just

south of the shelf break (East-box: cyan box in Fig. 5b, r$= 0.52$, lag$= 4$ days, BP$_{8D-10M}$). The positive sign indicates that strong eastward $\tau$ correlates with higher mCDW-temperatures at GC$_6$. During summer, the correlation is mostly insignificant.

     The temporal evolution of correlation between the mCDW-temperature and $\tau_{D18}$ averaged over the East-box shows that the correlation is significant for a large part of the mooring period and solely positive (mean or periods of significant correlation: r$= 0.56$, lag$= 3.7$ days, BP$_{8D-10M}$, Fig. 6a). The along-trough current's temporal evolution of correlation with the zonal $\tau_{D18}$

averaged over the ASP-box is also significant through most of the mooring period (Fig. 6b). The correlation is predominantly negative (mean of negative periods: $-0.54$, lag$= 1.8$ days, BP$_{8D-10M}$), but shorter periods of positive correlation occur (mean of





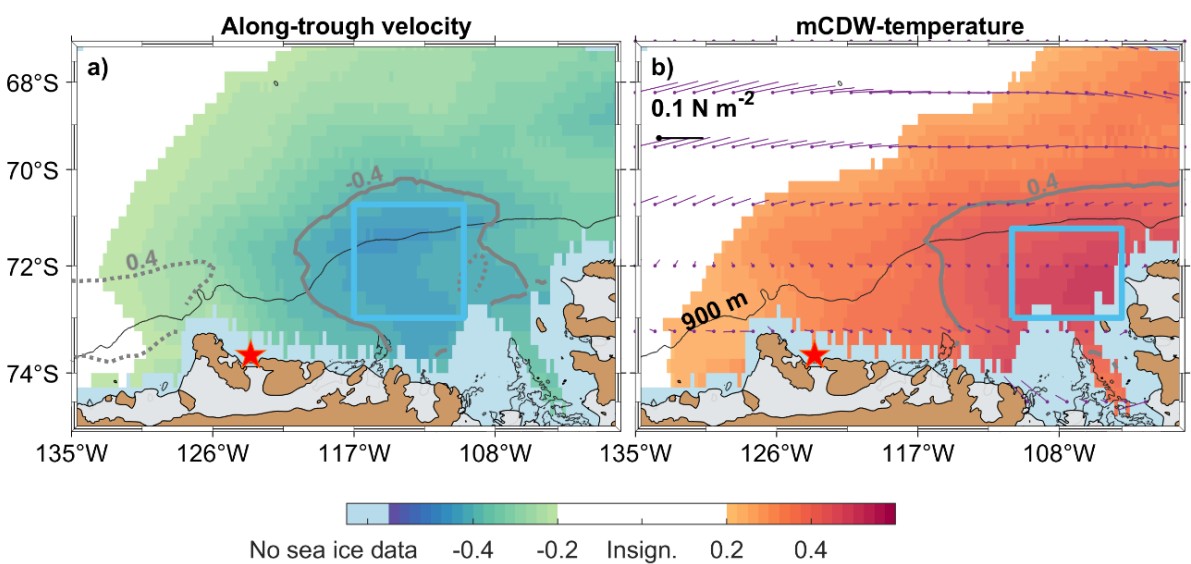

**Figure 5.** Spatial distribution of significant correlation during winter (colors) between zonal $\tau_{D18}$ and a) depth-averaged along-trough velocity at $GC_6$, and b) mCWD-temperature at $GC_6$. The correlation calculations are based on $BP_{8D-10M}$-filtered time-series averaged to daily values. Light blue regions along the coast indicate missing data on sea ice movement and white indicates insignificant correlation. Grey contours indicate a correlation of $\pm0.4$ for winter (solid) and summer (dashed). Cyan boxes mark the areas used for average $\tau_{D18}$ in Fig. 6. The red star and black contour mark $GC_6$ and the 900 m isobaths, respectively. In b), the purple lines with dots at their origin indicate mean $\tau_{D18}$ (scale in black). Values over land are removed.

positive periods: 0.49, lag= 0.7 days, $BP_{8D-10M}$), most notably during summer 2017. The occurrence of periods with positive correlation is independent of the parameterization of $\tau$ (Appendix B).

## 3.3 The regional model

We investigate the connections between the mCDW-temperature, the along-trough current, and $\tau$ using the regional model, with particular attention to how the $GC_6$-location is connected to other areas in the Amundsen Sea. First, we briefly describe hydrography and circulation in the model at the mooring site and the long-term atmospheric variability. We then look at large-scale variability of on-shelf temperature, and finally at the overall current and temperature's relation to $\tau_{ERA-I}$.

### 3.3.1 The long term state of the $GC_6$ region

The temperature output at $GC_{6\_mod}$ from the model over the period 2001-2017 indicates that $GC_6$ was deployed during a relatively cold period (i.e., deep $-1°C$ isotherm, Fig. 7c). This fits relatively well with the temperature indications from the available CTD profiles from 2000 and 2007 (Fig. 3a, Jacobs et al., 2013), and from 2016 and 2018: 2007 had a shallow $-1°C$ isotherm, in 2000 and 2018 it was deep, and in 2016 at medium depth (not shown). The match between the modeled and observed temperatures in 2014 is weaker: the seal-borne profiles show high bottom temperatures (Fig. 3a) and just a slightly





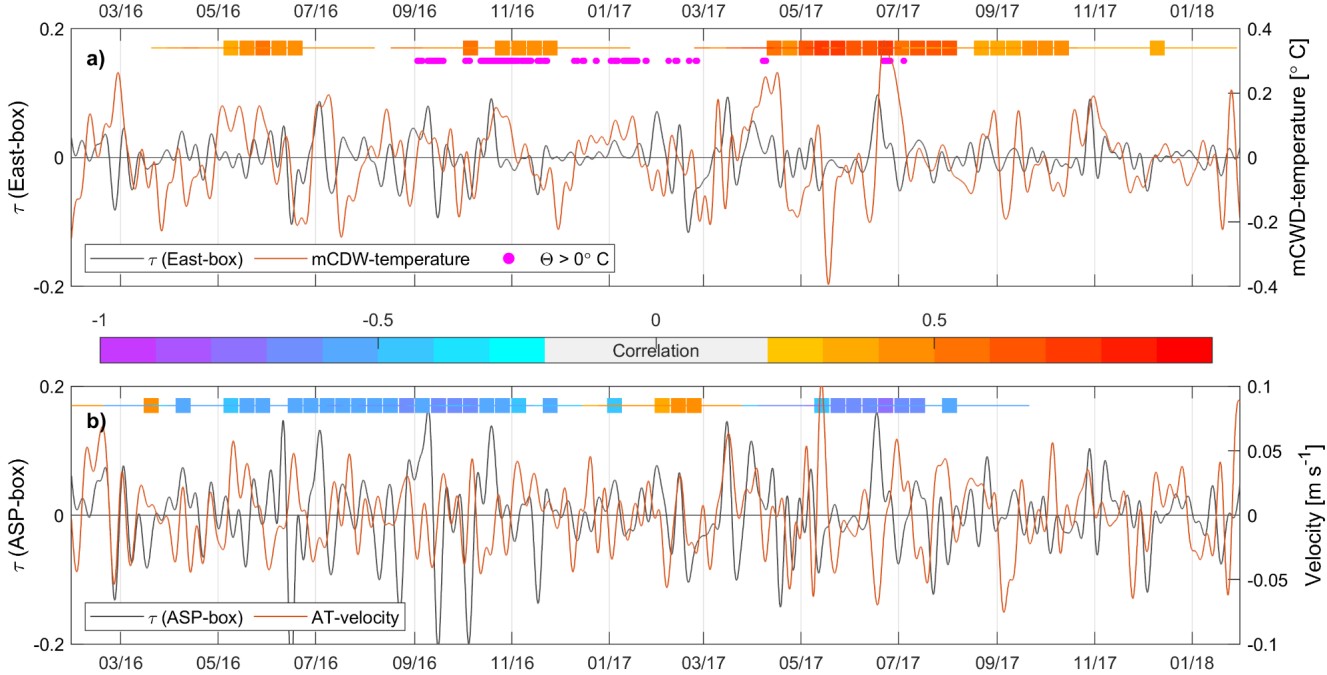

**Figure 6.** a) $\tau_{D18}$ (black line) averaged over the East-box and mCDW-temperature at the mooring site (red line). The correlation between the time-series is indicated by boxes and lines: Each colored box is the center of a 100-day window (lines) with significant correlation of magnitude given by the colorbar. The magenta dots show when temperatures above $0°$C are present. b) is analogous to a), but for $\tau$ averaged over the ASP-box and along-trough (AT) velocity instead of temperature. All time-series are filtered with $BP_{8D-10M}$).

deeper $-1°$C isotherm than in 2007 (not shown), contrary to the model which shows a strongly depressed $-1°$C isotherm in 2014. On average, the modelled $-1°$C isotherm at $GC_{6\_mod}$ is found at $342 \pm 48$ m depth (2001-2017). The difference between its shallowest and deepest periods is roughly 200 m and the amplitude of the low-frequency variability is larger than the variability of higher frequencies. (Fig. 7c). The average depth-mean rotated velocity at $GC_{6\_mod}$ at depths corresponding to the observed depths is $3.7 \pm 4$ cm s⁻¹ toward the ice shelf, similar to the observed average current.

The SIC over the $GC_6$-Trough, the wind field, and the wEK in the SB-box display large year-to-year variability (Fig. 7a). The wintertime SIC is always high, whereas it is highly variable between summers, ranging from $\sim 80\%$ to ice-free (Fig. 7a). The winds over the shelf break north of $GC_6$ are generally easterly in summer, as indicated by the zero-contour (Fig. 1c). There is no trend in the direction or strength in the average winds during 2001-2017, but the mooring period is within a period of relatively weak $\tau$ over the shelf break north of $GC_6$ (not shown). In some seasons, e.g., summer 2016 (winter 2011), the

zero-contour does not migrate north (south) (Fig. 7a). wEK is generally highest at the end of winter, and lowest at the end of summer, but shows strong positive anomalies during 2001-2003 and 2015-2017, which includes the mooring period (Fig. 7a,b).



**Figure 7.** Time-series of de-seasoned a) SIC at $GC_6$ (filled green), b) cumulative Ekman pumping anomaly, wEK, (ERA 5, filled blue) averaged over the SB-box (Fig. 1a), and c) the estimated monthly mean meridional position of the zero-contour (filled orange). The subscript 'SC' denotes the seasonal cycle. Light grey lines in a-c) are the estimated seasonal cycles, and dark grey, dark blue and dark orange in a-c) are the corresponding time-series (not de-seasoned). d) De-seasoned modelled depth of the $-1°C$ isotherms at $GC_{6\_mod}$ (purple), $V_{CN}$ (light purple), and $V_{NE}$ (pink). e) De-seasoned modeled depth of the $0°C$ isotherm (pink) and the along-flow (southeast) velocity at $V_{NE}$ (black). In c,d) thin lines are monthly means and thick lines are 12-month moving averages. In all panels, the time period with mooring measurements is marked by turquoise background.





### 3.3.2 Long-term variability in isotherm depth

The variability in isotherm depth is important both at the shelf break for admitting warm water onto the continental shelf and

on the continental shelf for the warm water's access to the base of the ice shelf. The evolution of the modeled isotherm depth anomalies (seasonal cycle removed and $LP_{8D}$) follows the main flow patterns on the continental shelf, as shown in the video in the supplementary material. Anomalies of deep and shallow isotherms in the $GC_6$-Trough seem primarily to originate from two regions: the trough northeast of $GC_6$ which is connected to the warm waters north of the shelf break, and along the coast from regions east of Carney Island. Some anomalies travel from the eastern Amundsen Sea around Bear Ridge and continue

westward along the coast (sketched arrows in Fig. 8d).

We select two locations in the regional model based on these pathways, one in the northeastern trough ($V_{NE}$) and one just north of Carney Island ($V_{CN}$, Fig. 8a), in order to better understand the pathways and time scales of isotherm depth anomalies traveling toward $GC_6$. By comparing the $-1°C$ and $0°C$ isotherm depths at $V_{NE}$ and $V_{CN}$ with $GC_{6\_mod}$ from 2001-2017, we note three relations: First, the isotherm depth at all three locations co-vary (Fig. 7c), with a stronger relationship between

$GC_{6\_mod}$ and $V_{CN}$ (thin lines, Fig. 7c). Peaks that occur at all three locations (e.g., in 2003) tend to first occur at $V_{NE}$, then at $V_{CN}$, and finally at $GC_6$, and have a lag of 1-3 months, which is comparable to the $\sim 3$ months advection timescale from the shelf break north of $GC_6$ to $GC_6$. Second, the low-frequency variability of the isotherm depths (12-month moving averages, Fig. 7c) responds to variations in SIC and wEK. High summer-time SIC and positive wEK are favorable for a thick warm layer the following winter through weak convection and lifted isotherms. Correspondingly, low summer-time SIC and negative wEK

favor a thin warm layer. The thermocline depth at $GC_{6\_mod}$ appears to be particularly sensitive to these fluctuations compared to $V_{NE}$ and $V_{CN}$ since the low-frequency variability of the $-1°C$ isotherm depth has the largest amplitude at $GC_{6\_mod}$ (Fig. 7c). The response time to variations in SIC and wEK is up to a year due to the slow deepening of the mixed layer after a summer of low SIC followed by sea ice formation (Fig. 7). This response time agrees with the observed time lag from the end of the main sea ice formation period in fall to the time when the $-1.8°$ isotherm reaches its deepest point at $GC_6$ (Fig. 4a,d). Third, at $V_{NE}$

the current strength and the thickness of the warm layer are correlated (r= 0.61, Fig. 7c), with a strong southeastward current corresponding to a thick warm layer.

### 3.3.3 Regional isotherm variability

The depth of the modeled $0°C$ isotherm averaged over 2016-2017 increases from about 200 m at the shelf break to 400 m depth in the $GC_6$-Trough. There is a similar increase from east to west (Fig. 8a), consistent with previous observations (Jacobs

et al., 2012) and with Ekman downwelling along the coast. The main warm period in the observational records from $GC_6$ ($\Theta > 0°C$: Sept. to Dec. 2016, Fig. 4a) lags a shallow $0°C$ isotherm over the overall continental shelf by several months (Fig. 8b). Comparing summers 2016 and 2017, preceding the warm and cold winter at $GC_6$ respectively, the isotherms in both the north-eastern trough and the Dotson-Getz Trough were shallower in 2016 than in 2017 (Fig. 8d). This delay between a generally shallow isotherm on the continental shelf and warm temperatures at $GC_6$ supports that advective processes ($\sim 3$

months) contribute to bringing heat to the mooring region.



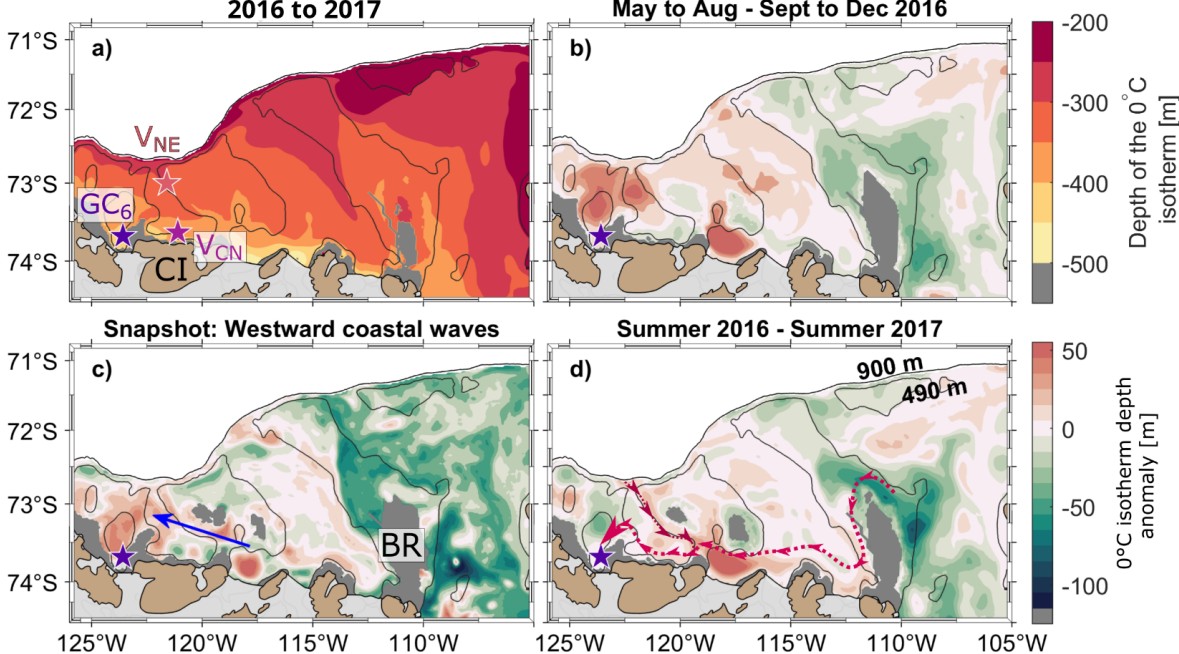

**Figure 8.** a) The mean depth of the 0°C isotherm during 2016-2017. Carney Island (CI), Bear Ridge (BR), the virtual mooring locations $V_{NE}$ (pink star) and $V_{CN}$ (light purple star), and $GC_6$ (dark purple) are marked in colors corresponding to Fig. 7b. b) The difference between May to August 2016 and September to December 2016. c) A snapshot from 8[th] of June 2016 showing an example of wave features along the northern coast of Carney Island (seasonal cycle removed and filtered with $LP_{8D}$). The blue arrow indicates the propagation direction. d) The difference between summer 2016 and summer 2017. The red dashed arrow indicates the suggested pathways of anomalies in isotherm depth. In all panels, the black contours indicate the 900 m and 490 m isobaths. The grey regions indicate that the deep 0°C isotherm is not present.

The propagation time scales of the anomalies, however, varies, likely impacted by the complex interactions of anomalies from the north and east that meet north of Carney Island. An anomaly that entered the northeastern trough in December 2016, for example, reached the $GC_6$-Trough already in January, while an anomaly in April 2017 arrived at the trough in June. The isotherm depth anomalies also reveal that eddies occasionally get "trapped" in the $GC_6$-trough, e.g., in June 2017, possibly helping sustain the warm peak (Fig. 4a). Occasionally, anomalies travel as waves westwards along the coast, visualized by the snapshot in Fig. 8c.

### 3.3.4 Spatial correlation of ocean surface stress with ocean currents and temperature

To further study the observed relationship between $\tau_{D18}$, the along-trough current, and mCDW-temperature we investigate how $\tau_{ERA-I}$ over the ASP-box and the East-box correlates with the daily modeled currents and the temperatures on the continental shelf, respectively (Fig. 9). Regions of strong currents generally have significant correlation with the ASP-stress (Fig. 9c,d). Three regions stand particularly out with positive correlation in the deep currents: the northeastern trough, the Dotson-Getz Trough, and the shelf break region stretching eastward to the entrance of the Dotson-Getz Trough. Positive correlation indi-





cates that a positive (eastward) ASP-stress anomaly leads to a positive current anomaly, i.e., enhanced current in the mean flow direction. At the $GC_6$-location, the positive correlation of ASP-stress with modeled currents disagrees with the negative

correlation with the observed current. As the model's representation of the current's short-term variability at $GC_6$ is unreliable and its local bathymetry is inaccurate, this is not surprising.

Above 100 m the correlation is negative along the coast north of Carney Island, i.e., westward (negative) $\tau_{\text{ERA-I}}$ enhances the westward current. Similar to below the $0°C$ isotherm, the correlation above 100 m is positive along the shelf break north of $GC_6$, and partly into the GD-Trough, but slightly weaker (Fig. 9d). This indicates that eastward $\tau_{\text{ERA-I}}$ induces an eastward

anomaly throughout the water column in these regions. In the regions of high correlation, the lag between $\tau_{\text{ERA-I}}$ and the surface currents and deep currents is 0-1 days and 1-2 days, respectively.

The correlation between $\tau_{\text{ERA-I}}$ and the $0°C$ isotherm depth (Fig. 9e) has one main similarity to the correlation with the deep currents: the correlation within the northeastern trough is positive and relatively strong. Eastward $\tau_{\text{ERA-I}}$ consequently induces both a shoaling of the $0°C$ isotherm depth and a strengthened southeastward current. This indicates that the long-term result of

correlation between current and isotherm variability from $V_{\text{NE}}$ holds for the entire northeastern trough. At $GC_6$ the correlation is also positive, in agreement with observations. As expected, above 100 m depth the correlation with temperature is mostly insignificant, which emphasizes that $\tau_{\text{ERA-I}}$ influence the ocean temperature mostly through deep-reaching dynamic processes.

We finally note that the overall correlation between $\tau$ and currents and temperature is strongest during winter, in agreement with results from $GC_6$ (Fig. 5). 2016 is also characterized by larger areas of significant and stronger positive correlation than

2017 (not shown), making 2016 similar to the winter average.

## 4 Discussion

### 4.1 Differences from the Siple and Dotson-Getz Troughs

The hydrographic conditions at $GC_6$ highlight the importance of bathymetry: despite the geographic proximity to the Siple and Dotson-Getz Troughs, the hydrography differs greatly (Fig. 3 and Assmann et al. 2019; Wåhlin et al. 2013). The variability has

been observed in snapshot CTD profiles reported in Jacobs et al. (2013). There are three fundamental hydrographic differences: i) The maximum temperature observed in the $GC_6$-Trough is $0.19°C$ (2014, seal-borne CTD, Fig. 3a), i.e., pure CDW is absent. The western GIS is the only region where the ice shelf base is potentially in contact with pure CDW, which crosses the $\sim 570$ m deep Siple Trough sill, and is regularly present at its ice front (Fig. 3, Assmann et al. 2019 their Fig. 2). The majority of this warm water is, however, likely too deep to interact directly with the ice shelf base (Assmann et al., 2019). Since $GC_6$ records

temperatures above $0°C$ and is moored on the slope of the trough, unmodified CDW could be present, but unobserved, in the deepest regions of the trough. However, the historical profiles record weak temperature gradients below the thermocline (Fig. 3a) – the mooring is likely representative for temperatures at depth. ii) mCDW influenced by meltwater is unobserved in the $GC_6$- and Siple Troughs, but is registered in the Dotson-Getz Trough (Wåhlin et al., 2010). While the Dotson-Getz Trough sill depth is $\sim 70$ m shallower than the Siple Trough sill, the shallower isotherms north and east on the continental shelf enable

basal melt. iii) Seasonality in deep currents and isotherms is absent in the $GC_6$- and Siple Troughs in 2016-2018, but is present





**Figure 9.** a,b) Bathymetry (color) and mean modeled currents (white sticks, 2016 to mid-2017) at every $7^{th}$ regional model grid point during winter a) below the $0^{\circ}$C isotherm, and b) above 100 m depth. The current scale is indicated in the top left corner, and the current is directed from the white circle indicating the grid cell center. Currents weaker than 1 cm s$^{-1}$ are omitted. The red star indicates GC$_6$. c,d) Maps of correlation between modelled currents and the zonal $\tau_{ERA-I}$ averaged over the ASP-box (cyan) c) below the $0^{\circ}$C isotherm and d) above 100 m depth. Grey regions have insignificant correlation, and the white region is outside the on-shelf study domain. Hatched regions have mean currents less than 1 cm s$^{-1}$. e,f) Are analog to panels c,d), but show correlation between the zonal $\tau_{ERA-I}$ averaged over the East-box (cyan) and e) the depth of the $0^{\circ}$C isotherm, and f) temperature above 100 m depth. The contours shown are the 900 m (thin blue line) and 490 m (thick, blue line) isobaths. The magenta arrow shows the direction of the mean $\tau_{ERA-I}$ over the boxes. In c,d,e,f) all time-series are de-seasoned and LP$_{8D}$ is applied, and the color of the star shows the correlation between $\tau_{ERA 5}$ and the along-trough current (c,d) and mCDW-temperature (e,f) from Fig. 5.




in the Dotson-Getz Trough (Wåhlin et al., 2013). Further south in the Dotson-Getz Trough the seasonality disappears (Jacobs et al., 2012), possibly due to mixing by internal waves and basin-scale eddies (Wåhlin et al., 2013).

The mean current at $GC_6$ brings an estimated $45\pm64\,\mathrm{MW\,m^{-1}}$ available heat toward the GIS front south in the $GC_6$-Trough, about one fifth of the values observed in the Siple Trough ($GW_6$). The heat transport at $GC_6$ might nonetheless be important
for the central GIS: The current in the Dotson-Getz Trough is even weaker and the heat transport conveyed toward the Dotson Ice Shelf is roughly half the value observed at $GC_6$ (Wåhlin et al., 2013), and still meltwater is observed here (Wåhlin et al., 2010). Estimates of heat transport based on single moorings, however, come with uncertainties related to capturing the width of the current and depend on the position of the core of the warm inflow relative to the mooring location. Estimates based on mooring arrays across the troughs would yield more reliable comparisons.

**4.2   Correlation between the along-trough current and $\tau$**

The heat transport at $GC_6$ is dominated by the variability in the along-trough current, as previously observed in the Dotson-Getz Trough (Wåhlin et al., 2013) and further east in the trough at $113°W$ (Assmann et al., 2013). The along-trough current's dominant response to $\tau$, however, where strong westward $\tau$ over the ASP region corresponds to a strong current toward the ice shelf ($BP_{8D\text{-}10M}$, Fig. 5a and 9b), differs from results from troughs both west (not shown) and east of the $GC_6$-trough. There,
the strongest currents toward the ice front are driven by eastward $\tau$ just north of the shelf break (Wåhlin et al., 2013; Assmann et al., 2013). This suggests that $\tau$ directly adjusts the barotropic component of the along-shelf currents, and consequently the direct flow of the undercurrent into these adjacent, deep troughs, while the relation between the variability at $GC_6$ and $\tau$ is more complex. The lag of 0-2 days between $\tau$ and the along-trough current, is, however, close to results from similar correlation analysis from other regions (Darelius et al., 2016; Wåhlin et al., 2013).

The location of the highest correlation between $\tau$ and the along-trough current (Fig. 5a) indicates that the wintertime link between $\tau$ and the along-trough current is related to ASP-specific features, such as the low wintertime SIC, the consistently strong winds that facilitate rapid momentum transfer from the atmosphere to the ocean, and its location over the coastal current. This suggests that current variability at $GC_6$ is connected to $\tau$-induced variability propagating with the coastal current. The non-significant correlation between $\tau_{\mathrm{ERA\text{-}I}}$ and the coastal current (Fig. 9c), which contradicts this hypothesis, might result
from the complex coastal geometry and bathymetry, large cyclonic systems, changes in the density structure which affects the balance between the barotropic and baroclinic components (Núñez-Riboni and Fahrbach, 2009; Kim et al., 2016), varying propagation speed of anomalies, irregular wave patterns along the coast, and trapped warm and cold anomalies. The variability in transport within the coastal current is also connected to the amount of meltwater produced from basal melt in ice shelf cavities (Nakayama et al., 2014a; Jourdain et al., 2017). We also note that the error in the estimation of $\tau$ introduced by
assuming a motionless ocean might influence this result in this region where the surface currents are strong.

In summer, the relationship between $\tau$ and the current at $GC_6$ shifts: the correlation turns positive, the region of highest correlation shifts west of Siple Island, and the northward component of the along-trough current increases. The increased northward component of the along-trough current during summer 2017 (Fig. 4c) reflects that the strong westward $\tau$ enhances the transport away from the ice shelf. The temporal and spatial changes in correlation are unlikely explained by momentum





transfer from sea ice, given the insensitivity of the correlation results to the different sea ice parameterizations for $\tau$ (Appendix B). However, the indirect effect of sea ice on the dynamics through, e.g., stratification, might be of importance.

### 4.3 Variability in heat content

Periods of increased temperatures at $GC_6$ are likely the result of at least two mechanisms. A local response: eastward $\tau$ associated with short-term Ekman upwelling and local lifting of the thermocline, and a long-term response: positive cumulative

Ekman pumping anomaly (wEK) high summertime SIC, and remote input of heat from the eastern Amundsen Sea and the deep ocean north of $GC_6$, superimposed on the local response, and adjusting the isotherms by up to 200 m (Fig. 7b). The local and long-term responses are distinguished by causing short (less than 3 weeks) and long (several months) periods of increased heat content, respectively.

The positive correlation between mCDW-temperature and $\tau_{D18}$ over the eastern shelf in winter ($BP_{8D-10M}$, Fig. 6b) is asso-

ciated with the local response, similar to observations in the Siple Trough, but different from the Dotson-Getz Trough where the high bottom temperatures are unrelated to the average wind over the shelf break (Wåhlin et al., 2013). This indicates that the bottom temperatures in the western Amundsen Sea are more sensitive to changing wind forcing than the eastern Amundsen Sea, which is supported by a higher standard deviation in isotherm depth west of the northeastern trough (not shown). During summer, $\tau$ has overall little influence on mCDW-temperature at $GC_6$, possibly a consequence of dominating westward winds, a

general depression of the thermocline at the shelf break, and increased stratification. A relation like the local response between eastward winds and increased on-shelf heat content has been shown in regions where the sill and troughs are deeper, such as in the Dotson-Getz Trough (e.g. Dotto et al., 2019), and the time scale of the local response is similar to under the Pine Island Ice Shelf (Davis et al., 2018). The cooling events at $GC_6$ that compare with those observed at $GW_{6F}$ (Steiger et al., 2021) might be triggered by strong winds over a region of low SIC east of Carney Island. The distance from this region to $GC_6$ is $\sim$ 150 km,

and the reduced temperatures occur roughly $3-5$ days after the increase in winds. The resulting propagation speed ($0.4-0.7$ m s$^{-1}$) matches the propagation speed from north of Siple Island to $GW_{6F}$ (Steiger et al., 2021), but the effect of the events on the heat content at $GC_6$ is less than at $GW_{6F}$.

The difference in heat content between 2016 and 2017 and the multi-yearly variability indicated by the model emphasizes the importance of the different atmospheric forcing between 2016 and 2017 specifically and the influence of the long-term

response in general. The two consecutive years (2015 and 2016) of southward shifted zero-contours might have induced the large positive wEK anomaly in 2016, possibly leading to shallower isotherms during summer 2016 than 2017 in both the coastal region and the northeastern trough (Fig. 8d). In 2017, the reduced wEK and strong thermohaline convection during freeze-up following the exceptionally low SIC might have caused the prolonged presence of WW below 300 m depth and deepening isotherms relative to 2016 (Fig. 4a). According to the long-term model results, however, both 2016 and 2017 were

relatively cold, despite the strong positive anomaly in wEK (Fig. 7a). Since an evident relation between wEK and heat content as observed in the Siple Trough (Assmann et al., 2019) is absent, this indicates that the low summertime SIC and relatively weak $\tau$ during the mooring period, as well as other large-scale atmospheric forcing mechanisms not assessed here, might be more central for heat content variability at $GC_6$ than wEK.





Several additional processes beyond the scope of this study govern shelf break regions. Passing coastally trapped waves (Chavanne et al., 2010) and the surface water thickness and composition (Daae et al., 2017) might affect the undercurrent's strength and depth. The undercurrent can further induce vortex systems and Rossby waves along the shelf break, bringing heat into troughs (St-Laurent et al., 2013). In the regional model, waves appear in the $0°C$ isotherm along most of the Amundsen Sea shelf break but dissolve in the region north of $GC_6$, suggesting a minor influence on the heat content variability at $GC_6$. We also note that the connection between increased basal melt from ice shelf cavities and the transport of the coastal current (Jourdain et al., 2017) might contribute to the variability we observe in the $GC_6$-Trough.

### 4.4 Large scale climate variability

Jacobs et al. (2013) suggested that the Amundsen Sea responds to large-scale changes as a unit, and consequently, the long-term variability at $GC_6$ could be influenced by far-field drivers such as the El Niño southern oscillation (ENSO) and anomalies in the Southern Annular Mode (SAM) (Dutrieux et al., 2014; Thompson et al., 2018; Spence et al., 2014). The future changes in ENSO are disputed (e.g. Perry et al., 2020), however, the SAM-index has a positive trend (a southward shifting zero-contour) due to $CO_2$ emissions and ozone depletion (e.g. Swart and Fyfe, 2012; Thompson and Solomon, 2002; McLandress et al., 2011). This might lead to reduced Ekman transport toward the coast, a relaxed ASF, and a combination of westward stress on the continental shelf and eastward stress along the shelf break, favoring increased heat transport toward the ice front at $GC_6$ and increased mCDW-layer thickness, respectively. While a clear relationship between the SAM-index and wEK and isotherm depth is absent on monthly time scales (not shown), the long-term trend might drive a slow change in the region. SIC also tends to be high during the positive mode of SAM (Lefebvre and Goosse, 2005), and indications of this occur over $GC_6$. The future state of SIC might be particularly important for the $GC_6$-Trough since isotherm deepening due to sea ice growth seems to have a larger impact here than elsewhere in the Amundsen Sea (Fig. 7c).

In the long term, the expected positive trend in SAM might influence the heat content at GC6 due to the link to atmospheric forcing. A permanently relaxed ASF would likely weaken the undercurrent and possibly enhance the relative importance of the wind-driven heat input from the east. However, the short-term relationship between $\tau$ and the bottom temperatures could be reduced by increased surface stratification due to increased sea ice melt. Still, the impact of future changes in SIC is uncertain given its contradicting response to positive SAM (Lefebvre and Goosse, 2005) and a warmer atmosphere. The shift in $\tau$ associated with the trend in SAM would not affect the katabatic winds, and thus the relation between $\tau$ and heat transport at $GC_6$ might be unchanged.

### 5 Summary and conclusions

This study provides a first detailed description of the hydrography, the ocean circulation, and its response to atmospheric forcing close to the front of Getz Ice Shelf (GIS) between Siple and Carney Islands using new mooring observations ($GC_6$) combined with output from a regional model (Nakayama et al., 2018, 2019) and historical CTD-profiles. The mooring data show temperatures over $−1°C$ throughout the mooring period and reoccurring periods with temperatures over $0°C$ (Fig. 4a).



The average heat transport is directed toward the ice shelf (Fig. 2d), but contrary to adjacent fronts (Assmann et al., 2019; Wåhlin et al., 2013) unmodified Circumpolar Deep Water (CDW) is absent (Fig. 3c). The data show no modification of CDW at depth by basal melt in the $GC_6$-Trough (Fig. 3c) as observed in the neighboring Dotson-Getz Trough (Wåhlin et al., 2010).

We analyzed the atmospheric drivers of the variability in the deep, warm temperatures and circulation at $GC_6$ and found a
link to ocean surface stress ($\tau$) on medium time scales (8 days to 10 months, Fig. 5 and 6). In winter, strong eastward $\tau$ over the eastern shelf break increases mCDW-temperatures, while strong westward $\tau$ over the Amundsen Sea Polynya (ASP) region strengthens the along-trough current. These relations agree with short-term relaxing of the Antarctic Slope Front (ASF), lifting of the thermocline and an accelerated undercurrent, and piling up of water along the coast and a strong westward barotropic current, respectively. Barotropic responses along the path of the coastal current may thus partly explain the high correlation and
short lag between $\tau$ over the ASP region and the along-trough current at $GC_6$, although this is not confirmed by the analyzed model fields. The temperature response to $\tau$ is similar in the $GC_6$- and Siple Troughs, while the response of the currents differs between the $GC_6$-Trough and the Siple- and Dotson-Getz Troughs, emphasizing the importance of bathymetry. The link between heat content, heat transport, and $\tau$ changes in summer – there is a shift in the dynamics induced by $\tau$ that is likely connected to stronger stratification and higher baroclinicity.

Winter conditions appear to favor wind-driven enhanced heat near the ice front (Fig. 6a). In winter, the wind field's zero-contour generally shifts southward, which we find to facilitate a warm $GC_6$-Trough. The positive wEK is also generally strongest in winter (Fig. 7b), in agreement with a suggested weakened ASF in mid-winter (Pauthenet et al., 2021). Uncharacteristic atmospheric forcing during the mooring period (Fig. 7a,b) likely explains the lack of seasonality in the $GC_6$ mooring record. Mixing by internal waves and basin-scale eddies might contribute, such as south in the Dotson-Getz trough (Wåhlin
et al., 2013). The warm 2016 was characterized by winter-like forcing, while the colder 2017 tended toward forcing more typical for summer (Fig. 4d,e). According to the long-term model results, however, the winter-like 2016 was not particularly warm (Fig. 7c). This emphasizes the complex interactions of forcing mechanisms, and strong positive wEK and a southward shifted zero-contour can be hindered from causing anomalously shallow isotherms on the continental shelf by counteracting forcing mechanisms such as low summertime SIC.

We conclude that although the entrance to the $GC_6$-Trough is sheltered from warm water inflow by bathymetry, it is connected to atmospheric forcing and conditions elsewhere on the continental shelf. This makes the ice front neighboring $GC_6$, just like the other ice fronts of GIS, vulnerable to future changes in the wind field, SIC, and thermocline characteristics. Detailed measurements of basal melt are needed to quantify the relative importance of this entrance. Sensitivity studies with a numerical model as well as improved bathymetry data at the fronts and under the ice shelf could provide further knowledge on how higher
temperatures and a shallower thermocline at $GC_6$ might affect meltwater production and the stability of GIS. This would help assess its contribution to future freshwater input toward the Ross Sea and how this could affect large-scale aspects such as the thermohaline circulation.



*Data availability.* Data from moorings GW$_6$ and GW$_{6F}$ are available through the NMDC data centre, and data from GC$_6$ are in the process of being published at the same location. Data from the S1 mooring are available at https://www.ncei.noaa.gov/access/metadata/landing-page/bin/iso?id=gov.noaa.nodc:0211128. The CTD data from tagged seals are available at https://www.meop.net/, and the ship-based CTD data taken onboard *N.B. Palmer* are available through the World Ocean Data Base and data from *Araon* in Lee (2016). The daily model output is available at https://ecco.jpl.nasa.gov/drive/files/ECCO2/LLC1080_REG_AMS/1080_run260_Elin, and the monthly model output at https://ecco.jpl.nasa.gov/drive/files/ECCO2/LLC1080_REG_AMS/run260/. New users must register for an Earthdata account at https://urs.earthdata.nasa.gov/users/new to access these files.

*Video supplement.* Movie S1

## Appendix A: Model validation

For the virtual mooring GC$_{6\,\text{mod}}$, we choose a location on the model grid slightly southwest of the actual mooring site as the depth at this location is more similar to the mooring depth. Trough openings are generally deeper and less restricted in the regional model bathymetry than in the IBCSO dataset. The entrance to the trough at the shelf break northeast of GC$_6$ is shallower than 490 m depth in IBCSO, closing off the connection to the open ocean, while the sill is deeper in the model bathymetry (Fig. A1a). IBCSO is expected to perform well in shelf break regions, and such features may thus help explain differences between model and observations.

Like similar models, such as the MITgcm setup used by Assmann et al. (2013), this regional model reports bottom temperatures that are too high and a warm layer that is too thick. In agreement with observations, however, CDW is not present at GC$_6$ Also, seasonality is imposed on the thinning (summer) and thickening (winter) of the warm layer that is not detected by GC$_6$. As the warm layer extends higher up in the water column in the model (about 100 m difference), the interaction between cool surface waters through deep ventilation and the warm layer seems to be more important in the model than in observations. Applying BP$_{8D\text{-}10M}$ to the $-1°$C isotherm, however, yield relatively good agreement between the model and observations (r$= 0.31$, Fig. A1b). Although the model does not resolve observed sudden changes in temperature as we observe in, e.g., May 2017, and imposes a few artificial peaks (Fig. A1b), the Fourier spectra support the conclusion that the magnitude of variability is relatively good both in the upper and lower layers of mooring extent (Fig. A1c).

Velocity variance ellipses from observations and model at the mooring location are similar in both magnitude and variability, and in variation with depth (Fig. 2d). This is true both for the model period as a whole and during most of the period when separated into monthly mean variance ellipsis. This indicates good agreement on the overall background state. The main difference is in the magnitude of the zonal component which tends to be of opposite sign, but as this is the minor component in both the model and observations, it is largely disregarded through rotation of the coordinate system along the direction of the mean flow (Fig. A1e). This slight difference in direction might be explained by differences in model and true bathymetry (Fig. A1a). Frequency spectra of velocity also indicate that the relative importance of variability on various time scales agrees well (not shown), and the depth profiles of decomposed principal components through EOF analysis are similar, though the


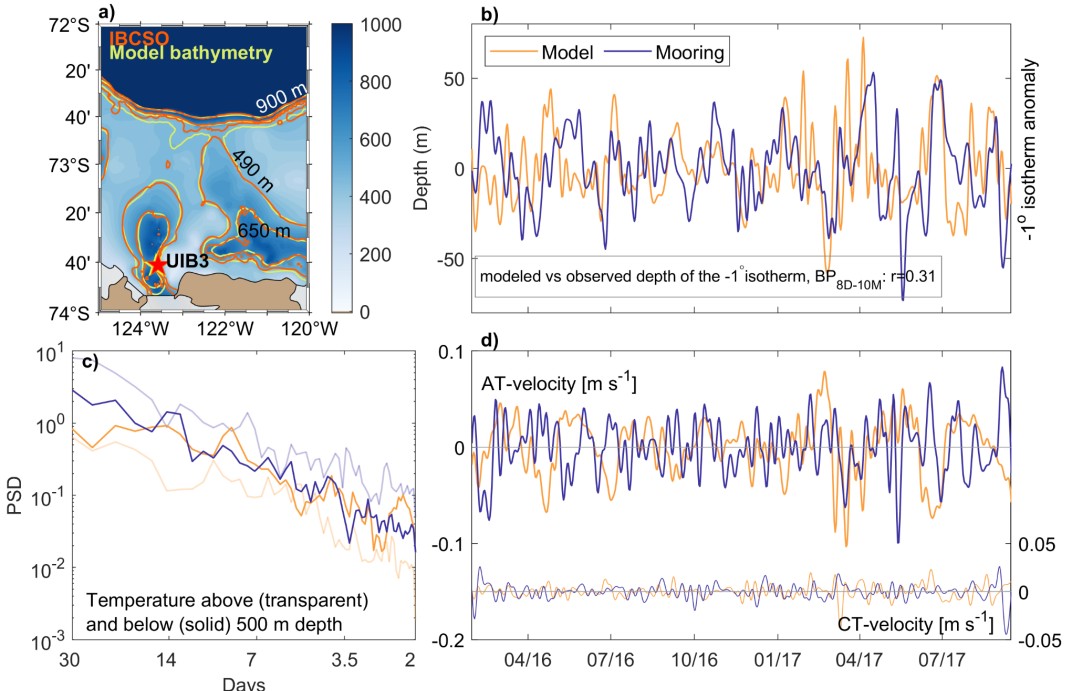

**Figure A1.** Comparison between a) selected isobaths from IBCSO (orange) and the model's bathymetry (yellow) (GC$_6$'s location: red star). 650 m is the mooring depth and at 490 m the model bathymetry indicates that the trough northeast of GC$_6$ is open, while it is closed in the IBCSO data. b) The anomaly of the $-1°C$ isotherm depth, c) the Fourier spectra of mean temperature above (transparent) and below (solid) 500 m depth, and the d) rotated along-trough current (AT-velocity, left axis) and cross-trough current (CT-velocity, right axis) at the mooring GC$_6$ (blue) and at the mooring location in the model (orange). Isotherm depth anomalies and rotated velocities are filtered with BP$_{8D-10M}$.

model overemphasizes PC1. The variability explained by the main components, PC1, and PC2, varies with time and co-vary for model and observations. These aspects all yield credibility to the use of the regional model to describe the average situation in the GC$_6$-Trough region on longer time scales.

There is, however, less agreement in velocity on shorter time scales (Fig. A1e). The velocity is represented well during specific periods (e.g., February-April 2017) and nearly opposite during other periods (e.g., November and December 2016).
Due to this, we do not rely on the short-term variability of the modeled velocity at the mooring site.

Just like in the daily model record, the monthly mean time-series (2001-2017) exhibit strong seasonality in temperature and salinity. The warm and saline layer at the bottom is thinner in summer than in winter – in summer the previous winter's deep ventilation pushes the deep layer of mCDW downwards. However, although ventilation of cool surface waters appears to affect and interact with the deep warm layer (Sect. 3.3.2), the model appears to underestimate the extent of the deep ventilation. WW
is never present below 200 m depth (the depth comparable to 300 m in the observations). In contrast, the mooring captures water below $-1.8°C$ down to 450m depth. Low salinity levels compared to the observations throughout the model period indicate that the brine release due to sea ice formation in fall is under-estimated in the model. This would explain the shallow extent of deep ventilation. The slope of the mixing line between WW and CDW in yearly mean TS-diagrams does however





increase (decrease) in league with decreasing (increasing) SIC (not shown). Extensive freeze-up after periods of low SIC means
deep ventilation of more saline waters, shifting the slope in the TS diagrams.

The depth of the modeled $0°$C isotherm averaged over 2016-2017 increases from about 200 m at the shelf break to 400 m
north of GC$_6$. There is a similar increase from east to west (Fig. 8a), which agrees with observations from Jacobs et al. (2012).
In the troughs, the isotherm stays shallow further onto the continental shelf.

## Appendix B: Calculations of ocean surface stress

For calculations that include $\tau$, we select three spatial boxes: one at the shelf break just north of GC$_6$, the "SB-box", one over
the Amundsen Sea Polynya and north to the shelf break, the "ASP-box" (see colored rectangles in Fig. 1a), and one on the
continental shelf further east, the "East-box". The first was chosen to explore if the shelf break processes are equally important
at GC$_6$ as at GW$_{6-7}$ further west (Assmann et al., 2019), and for estimations of wEK and the meridional position of the zero-
contour. The ASP-box was chosen since we find the highest correlation between $\tau_{D18}$ and the along-trough current at GC$_6$ is
over the ASP region (see Sect. 3.2). The East-box was chosen based on the region of highest correlation between $\tau_{D18}$ and the
mCDW-temperature at GC$_6$. We define the cumulative Ekman pumping anomaly (wEK) as the de-trended integral of $w_{EK}$ in
time. $dt$ is the time between observations and

$$w_{EK} = -\frac{1}{\rho}\frac{1}{\overline{f}}\frac{\Delta\tau^x}{\Delta y} \approx -\frac{1}{\rho}\left[\overline{\tau^x}\frac{\Delta f^{-1}}{\Delta y} + \frac{1}{\overline{f}}\frac{\Delta\tau^x}{\Delta y}\right] \approx -\frac{1}{\rho}\frac{\partial}{\partial y}\left(\frac{\tau^x}{f}\right) \approx \frac{1}{\rho}\left[\frac{\partial}{\partial x}\left(\frac{\tau^y}{f}\right) - \frac{\partial}{\partial y}\left(\frac{\tau^x}{f}\right)\right].\tag{B1}$$

In this approximation, the dependency on $\frac{\partial\tau^y}{\partial x}$ is neglected as the main gradients in $\tau$ are in the meridional direction, and $\frac{\Delta f^{-1}}{\Delta y}$
is neglected since $f$ is constant in longitude.

We include SIC and sea ice movement in the approximation of $\overrightarrow{\tau}$ to account for the drag of ice on the ocean following Dotto
et al. (2018):

$$\overrightarrow{\tau} = \alpha\overrightarrow{\tau}_{ice-water} + (1-\alpha)\overrightarrow{\tau}_{air-water}\tag{B2a}$$

$$\overrightarrow{\tau}_{ice-water} = \rho C_{iw}|\overrightarrow{U}_{ice}|\overrightarrow{U}_{ice}\tag{B2b}$$

$$\overrightarrow{\tau}_{air-water} = \rho_{air}C_d|\overrightarrow{U}_{air}|\overrightarrow{U}_{air},\tag{B2c}$$

where $\alpha$ is SIC, $C_{iw} = 5.50 \times 10^{-3}$ is the drag coefficient between ice and water, $\overrightarrow{U}_{ice}$ is the velocity of the ice, $\rho_{air} =$
$1.25$kg m$^{-3}$ is the density of air, $C_d = 1.25 \times 10^{-3}$ is the drag coefficient between air and water, and $\overrightarrow{U}_{air}$ is the 10 m wind.
Comparison of four differnet estimaes of $\tau$ using i) output from ERA 5, ii) Eq. B2a (Dotto et al., 2018), iii) Eq. B2c with $C_d$
parameterized using SIC (Andreas et al., 2010), and iv) Eq. B2c (only wind stress) shows that inclusion of sea ice reduces the
magnitude, but the variability is similar between all four estimations (Fig. B1). We do, however, assume a motionless ocean
and a spatially and temporally constant $C_d$, although it would be more accurate to use the relative velocities between air, sea




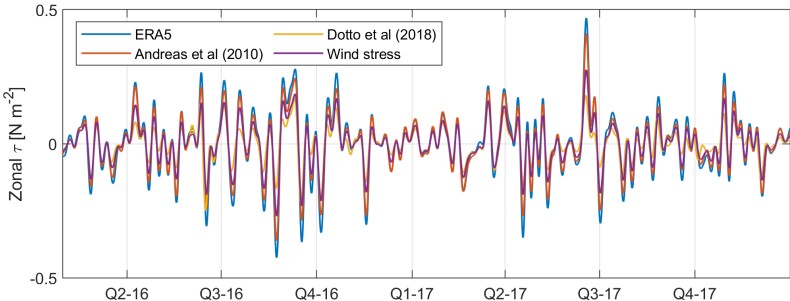

**Figure B1.** Different estimations of $\tau$ using the output from ERA 5 (blue), and explicit calculations using the parameterization of $C_d$ from Andreas et al. (2010) (red), parameterization of $C_d$ from Dotto et al. (2018) (yellow), and the wind stress without accounting for sea ice (purple). All timeseries are filtered with $BP_{\text{8D-10M}}$.

ice, and ocean, and $C_d$ as a function variables such as ridgedness, seasons, and geometry (Brenner et al., 2021). We choose to use Eq. B2a because previous studies have shown that the inclusion of sea ice and sea ice movement is important for a realistic estimate of momentum transfer into the ocean (Dotto et al., 2018), and do not have daily data on surface currents available for

the full study period.

    $\tau$ over the ASP-region appears to be particularly influential on the currents, compared to the rest of the Amundsen Sea area: High correlation occurs in a region similar to the ASP on a correlation map between spatially varying $\tau$ and the current at a fixed location at the shelf break where the correlation is high in Fig. 9c. The temporal evolution of correlation between heat transport and $\tau_{\text{D18}}$ is similar for stress averaged over the ASP and shelf break regions (SB-box), indicating that the large-scale

wind field induces the dominant current-variability at $GC_6$. Similarly, the four different parameterizations of sea ice for $\tau$ (Fig. B1) give similar temporal and spatial variability, but the magnitude of correlation increases with parameterizations including sea ice concentration and drift.

*Author contributions.* V.D. wrote the first draft, conducted analysis, and prepared the figures. E.D. processed the data. E.D., K.D., and N.S. improved the manuscript. Y.N. provided output from the regional model. E.D. and T.W.K. contributed to the field work. All authors read and

commented on the paper.

*Competing interests.* The authors declare no competing interests.

*Acknowledgements.* This work is supported by the Norwegian Research Council through project 267660 (TOBACO) and through the Korea Polar Research Institute, grant PE22110. The authors would like to thank Ilker Fer from the Geophysical Institute, University of Bergen, for comments on the manuscript and for lending instrumentation for $CG_6$.





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
