# Peer review of "Hydrography, circulation, and response to atmospheric forcing in the vicinity of the central Getz Ice Shelf, Amundsen Sea, Antarctica"

_Ocean Science, 2022_

## Author Comment (AC1)

**Seasonal cycle + high frequency signal with seasonal differences**

[Figure]

**Signal above bandpass filtered: 8D-10M**

[Figure]

**The bandpass filtered signal - the high frequency signal**

---

## Author Comment (AC2)

**Response to comments by reviewer nr 2 on "Hydrography, circulation, and response to atmospheric forcing in the vicinity of the central Getz Ice Shelf, Amundsen Sea, Antarctica" by V. Dundas, E. Darelius, K. Daae, N. Steiger, Y. Nakayama, and T. W. Kim (https://doi.org/10.5194/os-2022-13)**

We thank both reviewers for reading the manuscript thoroughly and for comments that have helped us make our main messages clearer. Below we respond to comments from reviewer nr 2. Our responses are in orange, and changes or additions to the manuscript are in italic. Line numbers refer to the new manuscript.

To indicate whether the lines we refer to have been changed since the first version of the manuscript we label the lines we refer to as CC (Changed for Clarity), AC (Added for Clarity), CR (Changed for Readability), or UC (UnChanged).
* * *
**Response to reviewer nr 2**

**Major comments**

**Comment nr 1:**

The authors did impressively thorough work, but I think not all of your work should be included in this manuscript. To my understanding, the key findings of this manuscript are the comparison between this ice-shelf front to other ice-shelf fronts nearby in the Amundsen Sea, the reason why they are different (from your manuscript, mainly bathymetry), and the mechanisms driving the temporal variation of the hydrographic properties (from your manuscript, mainly wind stress).

Re: Thank you for the positive and constructive feedback.

However, this manuscript focuses a lot on other minor details, e.g.,

**a)** "The correlation is predominantly negative, but shorter periods of positive correlation occur, most notably during summer 2017" (line 212), which is not related to any of your key findings

We have considered this and concluded that we want to keep this detail as it makes it clear that the main relations between ocean surface stress and currents/deep temperatures at GC6 cannot be generalized in time. We have, however, followed your advice and removed details concerning this, and the paragraph (line 192-199) now reads (CC):

"*The strongest correlation between the along-trough current at GC6 and zonal τ D18 occurs during winter over a region that roughly overlaps with the Amundsen Sea Polynya (ASP, Fig. 1a) and extends northward to the shelf break (Central-box, Fig. 5a, r= −0.49, lag= 0 days, BP8D-10M). The negative sign indicates that strong westward τ_D18 enhances the along-trough current toward the ice front. Only shorter periods of positive correlation, most notably during summer 2017, interrupt this pattern (Fig. 6b). The occurrence of periods with positive correlation is independent of the parameterization of τ (Appendix B).*

*For mCDW-temperature, the maximum correlation with τ_D18 is also found during winter, but in an area further east (East- box: cyan box in Fig. 5b, r= 0.52, lag= 4 days, BP8D-10M). The positive sign*

*indicates that strong eastward τ correlates with higher mCDW-temperatures at GC6 and holds for most of the mooring period (Fig. 6a).*"

In response to your comments, we have also elaborated slightly in the discussion (lines 356-361, CC/AC):

"*During summer, the correlation is mostly insignificant but shows signs of anti-correlation (Fig. 6a). We speculate that the insignificance is due to this shift from positive to negative correlation but that the shift happens too gradually for our moving windows to capture periods of significant negative correlation. We further suggest that the shift depends on the position of the zero-contour, as we observe that a northward shifted zero-contour coincides with periods of anti-correlation between the zonal stress and the mCDW-temperature. Dominating westward winds, general depression of the thermocline at the shelf break, and increased summer stratification also likely weaken the relationship between τ and the deep temperatures.*"

**b)** The green diamonds in Fig. 4abd for the strong cooling events found in GW6f, which are not significant in your mooring observations and not mentioned afterwards

Re: We have reduced the focus on these events since, as you comment, they do not have a major impact on GC6 in the end. However, since the events' amplitudes decay with distance from the coast, the effect on the hydrography closer to the ice shelf front is likely larger. This makes the events important for the ice shelf front, although they are less important for the mooring location itself.

Line 158-160 now reads (CC): "*There is also a strong variability on shorter time scales, for example, several abrupt cooling events in 2016 that correspond to cold events found at GW6F (Steiger et al. 2021, Fig. 4a,b,d, green diamonds), explained by wind-driven coastal trapped waves.*"

Line 362-367 now reads (CC/AC): "*The cooling events at GC6, which are similar to those observed at GW6F during the same period (Steiger et al., 2021), might be triggered by strong winds over a region of low SIC east of Carney Island. The estimated propagation speed of a coastal trapped wave from this region to GC6 (0.4 – 0.7 m s-1) matches the propagation speed of the coastal trapped wave observed at GW6F (Steiger et al., 2021). The effect of the events on the heat content at GC6 is less than at GW6F, possibly explained by the larger distance of GC6 from the ice shelf front. However, this implies that the signal is stronger at the ice shelf front than at GC6. Consequently, the effect of this wind-induced coastal wave is likely substantial at the ice shelf front in the GC6-Trough.*"

**c)** The authors also mentioned the advection time from the shelf break to the ice front several times, which is again, not related to any of your key findings

Re: We agree that this is not one of the main findings, and we have reduced the focus on this aspect. Since advection of heat from the eastern Amundsen Sea and the deep ocean north of GC6 contribute to the long-term variability in heat content at the GC6 mooring site, we still mention the time scale to relate it to these processes.

**Comment nr 2:**

I think the authors should focus on explaining the temporal variation of data with the physics behind it, instead of describing the variation. For example,

**a)** In lines 199-213, the authors describe every tiny anomaly in the line figures, without mentioning why the correlations are generally less significant in summer than in winter. Provided that no

significant seasonality in TS was observed in the mooring site, the seasonality of the significance of the correlations can be a fair scientific subject that is within the scope of this study

Re: See response to comment 1a.

**b)** In the model results section, the authors describe the shape of the lines in Fig. 7 panel by panel, without giving a convincing explanation (you hypothesize some mechanisms but do not explore further – you may also want to leave those hypotheses that lack evidence in future work) of how they are connected to the hydrography at the mooring site

Re: We have simplified this section and removed unnecessary details. Line 219-224 now reads (CC): "*The SIC over the GC6-Trough, the wind field, and the wEK in the SB-box display large year-to-year variability (Fig. 7a-c), with implications for the GC6 region. The wintertime SIC is always high, but highly variable between summers, ranging from ~80% to ice-free (Fig. 7a). There is no trend in the average wind velocity during 2001-2017, but the mooring period is within a period of relatively weak τ over the shelf break north of GC6 (not shown). Occasionally, the zero-contour does not migrate north (south) as expected during summer (winter) (Fig. 7c). wEK is generally highest at the end of winter, and lowest at the end of summer, but shows strong positive anomalies, including during the mooring period (Fig. 7a,b).*"

The reason for describing these differences comes on lines 237-244 and 368-377, which are unchanged since the first submission.

**Technical corrections**

Line 106-107, "We note that trough openings are generally deeper in the regional model than in the IBSCO bathymetry." I think the authors mean "deeper in the regional model than in the observations (e.g. multibeam survey)"? - as the model uses IBSCO?

Re: The model's bathymetry is *based* on IBCSO; it does not use it directly. To make this distinction clearer, we have rewritten lines 108-109 to (CC): "*We note that trough openings are generally deeper in the regional model's bathymetry, which is based on IBCSO, than in the IBCSO bathymetry itself.*"

Line 221, Fig. 7c, I think you mean Fig. 7d

Re: Corrected as suggested

Line 230, Fig. 7a, I think you mean Fig. 7a,b,c

Re: Corrected as suggested

Fig. 7d, in the legend, "current at V_NE" might be removed?

Re: Changed as suggested

Fig. 7e, (you might do it on purpose?) but the thick black line for the 12-month meaning average is missing

Re: Changed as suggested

Line 251, "Peaks that occur at all three locations tend to first occur at V_NE, then at V_CN, and finally at GC6..." I do not see that in the figure? For me, the most striking peak was in 2007, and the peaks arrived at different locations in the opposite sequence.

Re: We agree that this is not evident from Fig. 7, and as this is not a major point, we removed this sentence.

Line 355, the comma after "Ekman pumping anomaly" is missing

Re: Changed as suggested

---

## Author Comment (AC3)

**Response to comments by reviewer nr 1 on "Hydrography, circulation, and response to atmospheric forcing in the vicinity of the central Getz Ice Shelf, Amundsen Sea, Antarctica" by V. Dundas, E. Darelius, K. Daae, N. Steiger, Y. Nakayama, and T. W. Kim (https://doi.org/10.5194/os-2022-13)**

We thank both reviewers for reading the manuscript thoroughly and for comments that have helped us make our main messages clearer. Below we respond to comments from reviewer nr 1. Our responses are in orange, and changes or additions to the manuscript are in italic. Line numbers refer to the new manuscript.

To indicate whether the lines we refer to have been changed since the first version of the manuscript we label the lines we refer to as CC (Changed for Clarity), AC (Added for Clarity), CR (Changed for Readability), or UC (UnChanged).
* * *
**Response to reviewer nr 1**

**Major comments**

**Comment nr1:**

Dundas et al. combine mooring observations near the central Getz Ice Shelf with a regional model to study the processes that regulate time variability of heat transport toward the ice shelf. They show that atmospheric forcing (winds) and sea ice located over the wider continental shelf and shelf break of the Amundsen Sea regulate temperature and velocity changes at the front of the central Getz Ice Shelf. The manuscript is interesting and the statistical analysis mostly appropriate. However, after reading carefully, the message of the manuscript is not very clear (to me at least) How do velocity and temperature respond differently to winds and why?

Re: Thank you for a thorough and constructive review. Our main message is that non-local ocean surface stress is a major driver of variability in currents and hydrography in the relatively cold GC6-trough. The velocity and temperature respond differently to the ocean surface stress on mesoscale time scales (in this study reflecting 8 days to 10 months) because the mechanism behind their wind-driven variability is not the same. While westward ocean surface stress drives a southward current at GC6, westward ocean surface stress drives colder temperatures at GC6. Advection of heat by the southward current is consequently not the primary driver of mesoscale temperature variability at the mooring. We hypothesize that the current variability is connected to rapid wind-driven changes in the barotropic component of the currents on the continental shelf and along the coast. Regarding temperature, we hypothesize that wind-driven variability is associated with rapid thermocline adjustment.

We have adjusted the manuscript to emphasize these differences on

a) Lines 324-332 (CC, regarding the current): "*The location of the highest correlation between τ and the along-trough current (Fig. 5a) indicates that the wintertime link between τ and the along-trough current is related to ASP-specific features, such as the low wintertime SIC, the consistently strong winds that facilitate rapid momentum transfer from the atmosphere to the ocean, and its location over the coastal current. Without this polynya region of reduced SIC, which occasionally extends northward towards the shelf break, and the disturbance of*

*potential fast-ice, the wintertime relation between wind and current would possibly be much reduced. This suggests that current variability at GC6 is connected to τ -induced variability propagating with the coastal current. We speculate that the response might be largely barotropic. Westward winds increase the along-shore sea level and enhance the westward barotropic current along the coast, affecting the variability at GC6. This process would induce a negative correlation between the ocean surface stress and current at GC6, which is what we observe in winter.*"

b) Lines 345-346 (CC, regarding mCDW-temperature): "*Periods of increased temperatures at GC6 are likely the result of at least two mechanisms. One is a short-term response where eastward τ is associated with short-term Ekman upwelling and local lifting of the thermocline.*"

c) Lines 414-424 (CC): "*We analyzed the atmospheric drivers of the variability in the deep, warm temperatures and circulation at GC6 and found a link to ocean surface stress (τ) on medium time scales (8 days to 10 months, Fig. 5 and 6). In winter, strong eastward τ over the eastern shelf break increases mCDW-temperatures, while strong westward τ over the Amundsen Sea Polynya (ASP) region strengthens the along-trough current. These relations agree with i) short-term relaxing of the Antarctic Slope Front (ASF), lifting of the thermocline and an accelerated undercurrent, and ii) piling up of water along the coast and a strong westward barotropic current. Barotropic responses along the path of the coastal current may thus partly explain the high correlation and short lag between τ over the ASP region and the along-trough current at GC6. However, the analyzed model fields do not confirm this. The opposite sign of correlation of τ with the current and mCDW-temperature emphasize that advection of heat by the southward current is not the primary driver of temperature variability at the mooring on these time scales. The temperature response to τ is similar in the GC6- and Siple Troughs, while the response of the currents differs between the GC6-Trough and the Siple- and Dotson-Getz Troughs, emphasizing the importance of bathymetry.*"

**Comment nr 2:**

It is not clear how the wind stress regulates the ocean heat flux. Is it the wind over the Amundsen Polynya, at the shelf break, or over the eastern Amundsen Sea?

Re:

a) As observed at the mooring site, the variability in heat transport is determined by the variability in current strength rather than by the temperature or thickness of the mCDW layer (with only one mooring, we cannot account for variability in current width).

b) Our correlation analysis suggests that the currents at the mooring location respond strongest (during winter) to ocean surface stress in the region stretching from the ASP and north to the shelf break. We describe this connection on lines 192-194 (UC) "*The strongest correlation between the along-trough current at GC6 and zonal τ D18 occurs during winter over a region that roughly overlaps with the Amundsen Sea Polynya (ASP, Fig. 1a) and extends northward to the shelf break (Central-box, Fig. 5a, r= −0.49, lag= 0 days, BP8D-10M).* "

This hence suggests that the wind stress over this extended ASP region is a primary factor regulating the current and consequently the heat flux towards the ice front in the trough. The wind stress over the eastern Amundsen Sea modulates the thickness of the mCDW-layer at GC6, but the effect on the heat flux variability is secondary. We describe this on line 176 (UC): "*The variability in heat transport is dominated by current variability.*"

The wind over the region stretching from the Amundsen Sea Polynya to the shelf break has the highest impact on the current at GC6. We have adjusted our discussion of the reason for this on lines 324-332 to make this aspect clearer (see the response to comment 1a)

**Comment nr 3:**

What about sea ice?

Re: Interannual differences in summertime sea ice concentration appear to contribute to the long-term isotherm variability at the mooring site: low summertime SIC one year deepens the isotherms the following season. We discuss this on

a) Line 237-244 (UC): "*Second, the low-frequency variability of the isotherm depths (12-month moving averages, Fig. 7c) responds to variations in SIC and wEK. High summertime SIC and positive wEK are favorable for a thick warm layer the following winter through weak convection and lifted isotherms. Correspondingly, low summertime SIC and negative wEK favor a thin warm layer. The thermocline depth at GC6_mod appears to be particularly sensitive to these fluctuations compared to VNE and VCN since the low-frequency variability of the $-1^\circ C$ isotherm depth has the largest amplitude at GC6_mod (Fig. 7d). The response time to variations in SIC and wEK is up to a year due to the slow deepening of the mixed layer after a summer of low SIC followed by sea ice formation. This response time agrees with the observed time lag from the end of the main sea ice formation period in fall to when the $-1.8^\circ$ isotherm reaches its deepest point at GC6 (Fig. 4a,d).*"
b) Line 346-348 (CR): "*The second is a long-term response where positive cumulative Ekman pumping anomaly (wEK), high summertime SIC, and remote input of heat from the eastern Amundsen Sea and the deep ocean north of GC6 adjust the isotherms by up to 200 m (Fig. 7b).*"
c) Lines 375-377 (CR): "*The absence of an evident relation between wEK and heat content such as observed in the Siple Trough (Assmann et al., 2019) indicates that the low summertime SIC, as well as other large-scale atmospheric forcing mechanisms not assessed here, is more central for heat content variability at GC6 than wEK.*"

In addition, sea ice modulates ocean surface stress. After investigating the estimates of ocean surface stress with and without accounting for SIC and sea ice movement in the parameterizations (eq B2 in the manuscript), we find that the inclusion of sea ice is not crucial to describing the relationship between the winds and the currents at GC6. Instead, we find that wind stress describes most of the variability of ocean surface stress. We describe this on

d) lines 341-343 (UC): "*The temporal and spatial changes in correlation are unlikely explained by momentum transfer from sea ice, given the insensitivity of the correlation results to the different sea ice parameterizations for τ (Appendix B).*"
e) line 338-339 (UC): "*We also note that the error in the estimation of τ introduced by assuming a motionless ocean might influence results in this region where the surface currents are strong.*"

**Comment nr 4:**

In general the connection between winds and ocean heat flux needs to be better explained. For example, if the heat flux is dominated by the velocity, the authors could focus on the velocity.

Re: To clarify, we consistently describe the along-trough current rather than the heat transport throughout the manuscript, following lines 176-177 (AC): "*We, therefore, focus our following analysis on along-trough current variability rather than heat transport*."

We only come back to heat transport in the discussion and summary to remind the reader of the importance of the along-trough current for the heat transport in the trough on:

   a) Line 307-313 (UC): "*The mean current at GC6 brings an estimated 45±64 MW m−1 available heat toward the GIS front south in the GC6-Trough, about one fifth of the values observed in the Siple Trough (GW6). The heat transport at GC6 might nonetheless be important for the central GIS: The current in the Dotson-Getz Trough is even weaker, and the heat transport conveyed toward the Dotson Ice Shelf is roughly half the value observed at GC6 (Wåhlin et al., 2013), and still, meltwater is observed here (Wåhlin et al., 2010). Estimates of heat transport based on single moorings, however, come with uncertainties related to capturing the width of the current and depends on the position of the core of the warm inflow relative to the mooring location. Estimates based on mooring arrays across the troughs would yield more reliable comparisons.*"
   b) Line 410-412 (UC): "*The average heat transport is directed toward the ice shelf (Fig. 2d), but contrary to adjacent fronts (Assmann et al., 2019; Wåhlin et al., 2013), unmodified Circumpolar Deep Water (CDW) is absent (Fig. 3c)*"

**Comment nr 5:**

The connection between shelf break and coastal currents is very intriguing, but again it needs to be clearly stated since early in the manuscript.

*Re:* We are not quite certain how to interpret this comment, but we try to clarify below.

   a) Regarding a direct connection between the currents along the continental shelf break north of GC6 and the current variability at GC6: We are confident that this does not cause the main current variability at GC6 as i) the sill is shallow, ii) the regional model does mostly not indicate a current crossing the continental shelf here, and iii) the region of high correlation with ocean surface stress is found further east, where we define the Central-box.
   b) Regarding a connection with the current along the ice shelf fronts from the east of GC6: We hypothesize that the relationship we observe between the ocean surface stress over the Central-box (previously called ASP-box), which partly extends over the shelf break, and the along-trough current at GC6 might be influenced by stress-induced variability along the path of the coastal current. However, since we do not have data from the coastal current itself, we do not pursue or emphasize this possible connection. We rather focus on the possibility of a barotropic response induced by strong westward winds over the region of generally lower SIC than its adjacent regions, and how this can induce variability at GC6, east of the Central-box.

Because of these two explanations, we have tried to downplay the role of the coastal current for the variability at GC6.

**Comment nr 6:**

The title might also be changed to reflect the atmospheric forcing on heat flux toward the Getz Ice Shelf (also because you combine obs and modelling, not only moorings…).

Re: Following your suggestion we have renamed the manuscript to *"Hydrography, circulation and response to atmospheric forcing in the vicinity of the central Getz Ice Shelf, Amundsen Sea, Antarctica"*

**Minor Comments**

Line 56-61: is there any reference describing the undercurrent behaviour in the Getz region? If not, I would avoid including this in the Introduction. You could provide more details later on based on the model you are using.

Re: The undercurrent in the Getz region is described in, e.g., Walker et al., 2013, Assmann et al., 2013, and Dotto et al., 2019. It is not as prominent as other regions in Antarctica, but it is also observed in the Amundsen Sea.  We have rewritten lines 58-60 (CC) to *"The warm along-slope undercurrent (Walker et al., 2013; Assmann et al., 2013; Dotto et al., 2019) may therefore cross the deep Siple and Dotson-Getz Trough sills (∼570 and ∼500 m deep), but not the shallower GC6-Trough's sill (∼460 m deep)."*

Figure 2d: could specify in the legend that solid means obs and dashed means model?

Re: Changed as suggested

Line 107: do you mean the multibeam-based bathymetry? As the model is using IBCSO.

Re: The model's bathymetry is *based* on IBCSO; it does not use it directly. To make this distinction clearer, we have rewritten lines 108-109 (CC): "*We note that trough openings are generally deeper in the regional model's bathymetry, which is based on IBCSO, than in the IBCSO bathymetry itself.*"

Line 118-119: Can you say something more about what you mean by "since the sea-ice movement is expected to be small along the coast in winter, we assume little loss of information"? if sea ice motion is small, then the ocean surface stress is small if sea ice present and the wind stress is dampened.

Re: Values of sea ice motion are missing in the NSIDC dataset closest to the coast, but the sea ice in this region is likely fast-ice in winter. Wind over fast-ice will not lead to stress on the ocean at this location. Therefore, since we investigate the correlation between spatially varying ocean surface stress and currents and temperature at GC6, we expect that the correlation is zero where the ocean surface stress is zero. Also, we find that the very highest correlation is away from the coast. Because of this, we assume that the lack of data along the coast is not crucial for this specific analysis.  To make this clearer, lines 120-221 now read (CC): *"The sea ice motion dataset is incomplete near the coast. Since the highest correlation between the ocean surface stress and the currents and hydrography at GC6 is away from the coast, we assume that the lack of data is not crucial for our analysis."*

Line 122: can you specify what you mean by "surface stress"? is it the ocean surface stress (wind and sea ice) or wind stress?

Re: This is just wind stress. We use this when analyzing the model fields because this is what was used to force the regional model. But based on our comparison of ocean surface stress and wind stress, this should not make a major difference. This is now specified on lines 124-125 (CC): *"For analysis involving model output, we use daily instantaneous surface wind stress reanalysis output from ERA-Interim (referred to as $\tau_{ERA-I}$ hereafter) since this is used to force the regional model."*

Line 132-133: if you apply a band pass filter between 8 days and 10 months, the seasonal cycle is not going to be removed.

Re: With a cutoff at ten months, we expect to eliminate the seasonal cycle since a full seasonal cycle can't be resolved within ten months. Although the filtering removes the mean seasonal signal in the variables, we observe a seasonality in the observed variability and its response to variability in the forcing (on time scales shorter than seasonal). This is not a result of an unsuccessful removal of the mean seasonal signal. We illustrate this with an example in the attached figure where we show

the effect of our bandpass filter between 8 days to 10 months: panel 1 shows a seasonal cycle plus a high-frequency signal with seasonal differences in amplitude. Panel 2 shows the same signal after bandpass filtering. Finally, panel 3 shows the band passed signal minus the original high-frequency signal to illustrate that what is left after the bandpass filtering is the original high-frequency signal added onto the seasonal signal in panel 1.

[Figure]

Line 153: have you removed the seasonal cycle to all data? Or maybe here you are not removing the seasonal cycle. If not, please specify. Also, it would nice to clarify in the Methods whether you remove the seasonal cycle throughout the analysis or not.

Re: Here, we have not removed the seasonal cycle. We mostly do this for correlation analysis and in Fig. 7, where we focus on anomalies. To make this clearer, lines 133-136 now read (CC): *"To remove diurnal and seasonal signals from the mooring observations and model output in our correlation analysis and when stated specifically (e.g. in Fig. 7), we use two Butterworth filters. For the observations, we apply a bandpass filter from 8 days to 10 months (BP$_{8D-10M}$) which removes the seasonal cycle. Seasonal differences in variability on higher frequencies are, however, maintained."*

Line 169-170: I cannot see in Fig. 4b a tendency for different behaviour in 2016 and 2017. Could you maybe highlight in the text specific events of flow away from the ice shelf? Or maybe the number of "northward flow" episodes in 2016 and 2017 to see if there is a difference.

Re: Thank you for pointing this out. We followed up on your suggestion and estimated the percentage of northward flow during summer (31%) and winter (18%). However, after also

considering suggestions from reviewer nr 2 about removing some details from the manuscript, we decided to remove this aspect from the manuscript. We think this will help to make our main messages clear for the reader.

Figure 5: Why don't include 4 panels showing winter and summer correlations in different plots? I am also wondering why you make the distinction between summer and winter after removing the seasonal cycle? Can you clarify? Maybe the reason is that you have not removed the seasonal cycle.

Re: although we remove the seasonal cycle, the variability of the time series might still follow a different pattern of variability in summer and winter. There can still be a summer regime and a winter regime. The response to the comment concerning lines 133-136 also addresses this topic.

Line 200: the ASP box is mostly north of the polynya. I would change the name of this box and be clear in the text the you are doing correlations with shelf break surface stresses and not with the polynya region.

Re: There is an offset between the ASP-box and the indicated ASP-region, and we acknowledge that there should perhaps be more overlap to name the box the "ASP-box". Following your advice, we have changed the name of the ASP-box to the Central-box. We will keep the outline of the ASP in Fig. 1 and keep the focus on the fact that although the very highest correlation is along the shelf break, the high-correlation region stretches down to the coast where the ASP is. This way we keep the connection to the ASP and the coastal current but bend the focus a bit more towards the shelf-break.

We have rephrased lines 192-194 (CC): "*The strongest correlation between the along-trough current at GC6 and zonal τ D18 occurs during winter over a region that roughly overlaps with the Amundsen Sea Polynya (ASP, Fig. 1a) and extends northward to the shelf break (Central-box, Fig. 5a, r= −0.49, lag= 0 days, BP8D-10M).*"

Line 207-213: Could you please clarify in the text (or better in the Methods) how you calculate time changes of the correlation?

Re: We address this on lines 128-132 (UC). "*To estimate the temporal evolution of correlation, we use a 100-day moving window with 10 days overlap. All correlation values are significant on the 95% level, with significance calculated following Sciremammano (1979). We allow a maximum lag of 7 days for correlation analyses, encompassing rapid barotropic responses but leaving out slow advective signals. The mooring record length and low velocities in the GC6-Trough yield too few degrees of freedom to allow for a lag on the order of the advection timescale of roughly three months from the shelf break to GC6.*"

Section 3.1 and 3.2.: It would be nice at the end of these two sections to summarize the main results from observations. In this way the reader can better connect observations with the modelling results in the following section.

Re: We have included a small summary at the beginning of section 3.3 (line 201-205, AC): "*From the observations, we found that the GC6-Trough is relatively cold compared to its neighboring troughs (Fig. 3), but that temperatures higher than 0◦C are present occasionally (Fig. 4a). Variability in the along-trough current and the mCDW-temperature are both driven by τ. However, while strong currents towards the ice shelf are generally driven by westward τ over a region stretching from the ASP to the shelf break (Fig. 5a), high mCDW-temperatures are generally driven by eastward stress further east on the continental shelf (Fig. 5b).*"

Line 260: do you mean Fig. 7e?

Re: Corrected as suggested

Section 3.3.3: I think that based on the analysis reported by the authors it is difficult to make strong conclusions on advection/waves. I would stress that you see coherent changes between the mooring location and the continental shelf with a few month lag, suggesting advection. Plus you can say few words on waves as potential mechanism, give also the recent results from Steiger et al. (2021), but no definitive conclusions (probably more appropriate for the Discussion).

Re: We have changed lines 254-257 (CC) to "*The propagation time scale of the isotherm depth anomalies vary, possibly impacted by the complex interactions of anomalies from the north and east that meet north of Carney Island. The isotherm depth anomalies also reveal that eddies occasionally get "trapped" in the GC6 trough, e.g., in June 2017, possibly contributing to sustain warm peaks (Fig. 4a). Occasionally, anomalies appear to travel as waves westwards along the coast, visualized by the snapshot in Fig. 8c.*"

Line 287: westward wind anomalies causing westward current anomalies shouldn't imply a positive correlation?

Re: It is correct as stated in the manuscript: The correlation should be negative because the sign of the wind stress is negative westward. In contrast, the sign of the westward current at this location is positive (due to rotation with the mean current direction). Lines 274-279 now reads (CC): "*The shallow currents (above 100 m) also have positive correlation along the shelf break north of GC6 and partly into the Dotson-Getz Trough, comparable to the deep currents but slightly weaker. This indicates that eastward τ ERA-I induces a positive anomaly in the mean current direction throughout the water column in these regions, except for the coast north of Carney Island. There, a negative correlation indicates that westward (negative) τ ERA-I enhances the westward current (positive, due to the local rotation of the coordinate system with the mean current direction). In the regions of high correlation, the lag between τ ERA-I and the surface currents and deep currents is 0-1 days and 1-2 days, respectively.*"

Line 312-315: The absence of meltwater at $GC_6$ does not imply that MCDW cannot melt the ice shelf as the meltwater outflow is presumably located somewhere else. I would say something along those lines here.

Re: We have been hesitant to include comments on the absence of meltwater and where it exits as we cannot know. However, following this comment, we have decided to include these lines (301-303, AC): "*We note that although meltwater is unobserved in the GC6-Trough, the heat available could still induce basal melt without being detected in our instrument set-up. The available data is insufficient to assess whether meltwater exits at shallow depths or through pathways underneath the ice shelf.*"

Section 4.2. I like this section that summarizes the connection between surface stress and ocean current. Just it is hard to understand how a negative correlation in winter between surface stress and current can arise. I would try at least to provide some speculation regarding this correlation.

Re: We speculate that at GC6, the westward winds increase the along-shore sea level and set up a westward barotropic current along the coast, causing variability at GC6. In this case, the variability in current at GC6 is connected to variability in the coastal current, and the correlation with the ocean surface stress is negative. We have added lines 329-332 to describe this (AC): "*We speculate that the response might be largely barotropic: Westward winds increase the along-shore sea level and enhance the westward barotropic current along the coast, affecting the variability at GC6. This*

*process would induce a negative correlation between the ocean surface stress and current at GC6, which is what we observe in winter.*"

Line 352-356: Please rephrease this sentence as it is hard to follow. You might need to divide it into two sentences.

Re: Line 345-348 now reads (CR): "*Periods of increased temperatures at GC6 are likely the result of at least two mechanisms. One is a short-term response where eastward τ is associated with short-term Ekman upwelling and local lifting of the thermocline. The second is a long-term response where positive cumulative Ekman pumping anomaly (wEK), high summertime SIC, and remote input of heat from the eastern Amundsen Sea and the deep ocean north of GC6 adjust the isotherms by up to 200 m (Fig. 7b).*"

Line 359: do you mean Fig. 6a?

Re: We have corrected this to "Fig. 5b."

Line 360: I would not call this "local response" as the wind forcing is located on the other side of the Amundsen Sea.

Re: We have changed this from a "local response" to a "short-term response", which is more precise.

Line 368-372: which type of waves are you referring to? Please specify.

Re: We are referring to coastal trapped waves. Lines 362-366 now read (CC/AC), "*The cooling events at GC6, which are similar to those observed at GW6F during the same period (Steiger et al., 2021), might be triggered by strong winds over a region of low SIC east of Carney Island. The estimated propagation speed of a coastal trapped wave from this region to GC6 (0.4 – 0.7 m s-1) matches the propagation speed of the coastal trapped wave observed at GW6F (Steiger et al., 2021). However, the effect of the events on the heat content at GC6 is less than at GW6F, possibly explained by the larger distance of GC6 from the ice shelf front.*"

Line 420: I would rephrase "medium time scale".

Re: We instead refer to mesoscale variability, and lines 414-415 now read (CC): "*We analyzed the atmospheric drivers of the mesoscale (8 days to 10 months) variability in the deep, warm temperatures and circulation at GC6 and found a link to ocean surface stress (τ, Fig. 5 and 6).*"

---

## Author Response (AR2)

**Response to reviewers' comments on "Hydrography, circulation, and response to atmospheric forcing in the vicinity of the central Getz Ice Shelf, Amundsen Sea, Antarctica" by V. Dundas, E. Darelius, K. Daae, N. Steiger, Y. Nakayama, and T. W. Kim (https://doi.org/10.5194/os-2022-13)**

We thank both reviewers for reading the revised manuscript and for providing technical corrections as well as suggestions for improvements. Below we respond to comments from the reviewers. Our responses are in orange. Line numbers refer to the new manuscript.
* * *
**Response to reviewer nr 1**

line 12, you cannot prove it (and I do not think you want to do it in this manuscript), so it should be "possibly due to anomalously…"

Re: Changed as suggested

line 166, "Fig. 4d" might be "Fig. 4b"?

Re: Corrected as suggested

lines 203-205, you can only show that they are correlated, but the correlation might not lead to causation. I suggest simply changing it to "However, while strong currents towards the ice shelf generally correspond to westward τ over a region stretching from the ASP to the shelf break (Fig. 5a), high mCDW-temperatures generally correspond to eastward stress further east on the continental shelf (Fig. 5b)."

Re: Changed as suggested

line 256, "possibly contribution to sustain" should be "possibly contributing to sustaining"

Re: Corrected as suggested

lines 317-219, I find this sentence difficult to follow. Do you mean something like "The along-trough current's dominant response to τ, however, differs from results from troughs both west and east of the GC6-trough: the strong westward τ over the continental shelf between the ASP and the shelf break corresponds to a strong current toward the ice shelf (BP8D-10M, Fig. 5a and 9b)"?

Re: Changed as suggested

Fig. 3 caption, the last line, should be "Circumpolar Deep Water" with capital initials for water mass

Re: Corrected as suggested

**Response to reviewer nr 2**

Line 38: I would add a reference to Dotto et al., 2020 (jgro) where the authors show mooring observations in the Dotson-Getz Trough.

Re: Added as suggested

Line 340-341: where is the summer correlation with winds west of Siple Island shown. Maybe you could add "not shown" if no figure highlights this.

Re: The contour of the 0.4 correlation coefficient during summer is indicated with a dashed grey contour in Fig. 5. For clarification, lines 340-341 now reads: "*In summer, the relationship between τ and the current at GC6 shifts: the correlation turns positive, and the region of highest correlation shifts west of Siple Island (dashed contour in Fig. 5a).*"